# CONTINUOUS-DISCRETE CONVOLUTION FOR GEOMETRY-SEQUENCE MODELING IN PROTEINS

**Hehe Fan[1,3]**∗**, Zhangyang Wang[2], Yi Yang[1] & Mohan Kankanhalli[3]**
[1]Zhejiang University
[2]The University of Texas at Austin
[3]National University of Singapore

## ABSTRACT

The structure of proteins involves 3D geometry of amino acid coordinates and 1D sequence of peptide chains. The 3D structure exhibits irregularity because amino acids are distributed unevenly in Euclidean space and their coordinates are *continuous* variables. In contrast, the 1D structure is regular because amino acids are arranged uniformly in the chains and their sequential positions (orders) are *discrete* variables. Moreover, geometric coordinates and sequential orders are in two types of spaces and their units of length are incompatible. These inconsistencies make it challenging to capture the 3D and 1D structures while avoiding the impact of sequence and geometry modeling on each other. This paper proposes a Continuous-Discrete Convolution (CDConv) that uses irregular and regular approaches to model the geometry and sequence structures, respectively. Specifically, CDConv employs independent learnable weights for different regular sequential displacements but directly encodes geometric displacements due to their irregularity. In this way, CDConv significantly improves protein modeling by reducing the impact of geometric irregularity on sequence modeling. Extensive experiments on a range of tasks, including protein fold classification, enzyme reaction classification, gene ontology term prediction and enzyme commission number prediction, demonstrate the effectiveness of the proposed CDConv.

## 1  INTRODUCTION

Proteins are large biomolecules and are essential for life. Understanding their function is significant for life sciences. However, it usually requires enormous experimental efforts (Wüthrich, 2001; Jaskolski et al., 2014; Bai et al., 2015; Thompson et al., 2020) to find out their function. Recently, with the development of deep learning, emerging computational and data-driven approaches are particularly useful for efficient protein understanding (Derevyanko et al., 2018; Ingraham et al., 2019; Strokach et al., 2020; Cao et al., 2021; Jing et al., 2021; Jumper et al., 2021; Shanehsazzadeh et al., 2020), including protein design, structure classification, model quality assessment, function prediction, *etc*. Because the function of proteins is based on their structure, accurately modeling protein structures can facilitate a mechanistic understanding of their function to life.

Proteins are made up of different amino acids. There are 20 types of amino acids (residues) commonly found in plants and animals and a typical protein is made up of 300 or more amino acids. Because these amino acids are linked by peptide bonds and form a chain (shown in Fig. 1), proteins exhibit a 1D sequence structure. Moreover, because amino acids are arranged uniformly in the chains and their orders are discrete, the sequence structure is regular. In this case, 1D Convolutional Neural Network (CNN) (Kulmanov et al., 2018; Hou et al., 2018; Kulmanov & Hoehndorf, 2021) and Long Short-Term Memory (LSTM) (Bepler & Berger, 2019; Rao et al., 2019; Alley et al., 2019; Strodthoff et al., 2020) are widely used to model the regular 1D sequence structure of proteins.

In addition to the 1D sequential order in peptide chains, each amino acid is with a 3D coordinate that specifies its spatial position in a protein. As shown in Fig. 1, those 3D coordinates describe a geometry structure, which is also crucial for protein recognition. As mentioned by Alexander et al. (2009),

---

∗Part of this work was done when Hehe Fan was a Research Fellow at National University of Singapore.

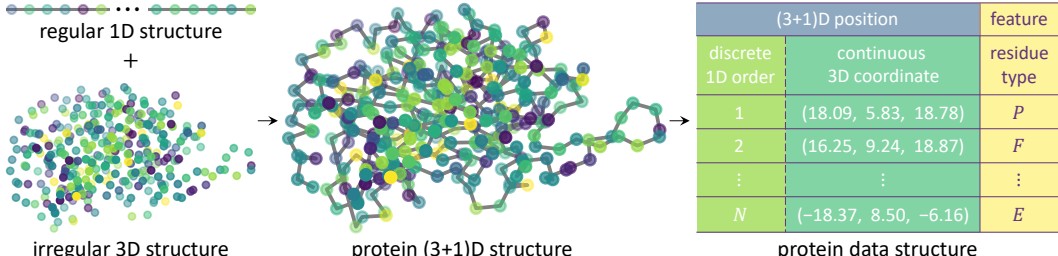

Figure 1: Illustration of the geometry-sequence structure of a protein. The dot color indicates amino acid (residue) types. 1) Amino acids are linked by peptide bonds and form a chain, which exhibits a regular 1D sequence structure because they are arranged uniformly and their sequential orders are discrete. 2) In addition, amino acids are with 3D coordinates that determine a geometry structure, which exhibits irregularity because they are distributed unevenly in Euclidean space and their coordinates are continuous variables.

proteins with similar peptide chains may fold into very different 3D geometry structures. Conversely, proteins with similar 3D geometry structures may have entirely different amino acid chains (Agrawal & Kishan, 2001). Therefore, it is necessary to consider both the 1D and 3D structures in protein modeling. However, different from the sequence structure, the geometry structure is irregular because amino acids are distributed unevenly in Euclidean space and their coordinates are continuous variables. This increases the challenge for neural networks to understand proteins.

To model the 1D and 3D structures in proteins, Gligorijević et al. (2021) employed an LSTM and a Graph Convolutional Network (GCN) for sequence and geometry modeling, respectively. Because the two structures are processed separately, it may not properly understand proteins' local geometry-sequence structure. In contrast, a few unified networks try to model the two types of structures together (Baldassarre et al., 2021; Jing et al., 2021; Zhang et al., 2022; Hermosilla & Ropinski, 2022). However, those methods process geometric and sequential displacements together or model the sequence structure in a similar way to geometry modeling, thus neglecting the regularity of the 1D structure. Moreover, because the length units of the sequential and geometric spaces in proteins are not compatible, treating their distances similarly may mislead deep neural networks.

In this paper, we first propose and formulate a new class of convolution operation, named Continuous-Discrete Convolution (CDConv), to make the most of the dual discrete and continuous nature of the data to avoid the impact of regular and irregular modeling on each other. Specifically, CDConv employs independent learnable weights to reflect different regular and discrete displacements but directly encodes continuous displacements due to their irregularity. Then, we implement a (3+1)D CDConv and use this operation to construct a hierarchical (3+1)D CNN for geometry-sequence modeling in proteins. We apply our network to protein fold classification, enzyme reaction classification, gene ontology term prediction and enzyme commission number prediction. Experimental results demonstrate the effectiveness of the proposed method. The contributions of this paper are fourfold:

- We propose a new class of convolution, *i.e.*, CDConv, which unifies continuous and discrete convolutions and makes the most of the regularity and irregularity in data, respectively.
- We implement a (3+1)D CDConv for geometry-sequence modeling in proteins. Based on the convolution, we construct deep neural networks for protein representation learning.
- We conduct extensive experiments on four tasks and the proposed method surpasses the previous methods by large margins, resulting in the new state-of-the-art accuracy.
- We find that amino acids in central regions may be more important than those in surface areas. This may need to be verified via biochemical experiments in the future.

## 2   RELATED WORK

**Protein Representation Learning.** Protein representation learning attracts increasing attention in the fields of protein modeling and structural bioinformatics and is critical in biology. Because proteins are sequences of amino acids, 1D CNN, LSTM and Transformer are employed for sequence-based protein representation learning (Amidi et al., 2018; Kulmanov et al., 2018; Hou et al., 2018; Rao et al.,

2019; Bepler & Berger, 2019; Alley et al., 2019; Strodthoff et al., 2020; Shanehsazzadeh et al., 2020; Kulmanov & Hoehndorf, 2021). Besides the sequence structure, the 3D geometry structure is also used to enhance protein representations, based on amino acid or atom 3D coordinates (Derevyanko et al., 2018; Ingraham et al., 2019; Gligorijević et al., 2021; Baldassarre et al., 2021; Jing et al., 2021; Wang et al., 2021; Hermosilla et al., 2021; Zhang et al., 2022; Chen et al., 2022; Hermosilla & Ropinski, 2022) or protein surfaces (Gainza et al., 2020; Sverrisson et al., 2021; Dallago et al., 2021). Our method is based on 1D sequential orders and 3D geometric coordinates of amino acids. Different from those existing methods, we propose to employ discrete and continuous manners to model regular orders and irregular coordinates, respectively.

**Discrete and Continuous Convolution.** Convolution is one of the most significant operations in the deep learning field and has made impressive achievements in many areas, including but not limited to computer vision and natural language processing. Convolution can be defined as functions on a discrete or continuous space. For example, images can be seen in a 2D discrete space of pixels and thus a number of 2D discrete CNNs are used for image processing (Krizhevsky et al., 2012; Simonyan & Zisserman, 2015; Szegedy et al., 2016; He et al., 2016). Similarly, languages can be seen in a 1D discrete space of words or characters and therefore can be modeled by 1D discrete CNNs (Kim, 2014; Yin et al., 2017; Bai et al., 2018). Continuous convolutions are usually used to process irregular data (Schütt et al., 2017; Simonovsky & Komodakis, 2017; Wang et al., 2018; Romero et al., 2022), especially to 3D point clouds (Thomas et al., 2018; Wu et al., 2019; Hu et al., 2020; Fuchs et al., 2020; Fan et al., 2021; Shi et al., 2021), or sometimes used to improve flexibility for discrete convolutions (Shocher et al., 2020). When implementing an instance of the proposed (3+1)D CDConv for protein modeling, the 3D processing part is inspired by those point cloud methods.

## 3 PROPOSED CONTINUOUS-DISCRETE CONVOLUTION

In this section, we first briefly review the discrete and continuous convolutions for regular and irregular structure modeling, respectively. Then, we present the proposed Continuous-Discrete Convolution (CDConv) in detail. Third, we introduce the (3+1)D CDConv for geometry-sequence modeling in proteins. Fourth, we present an implementation for the proposed (3+1)D CDConv. Last, we incorporate our (3+1)D CDConv into a hierarchical architecture for protein representation learning.

### 3.1 PRELIMINARY: CONVOLUTION

The power of convolutional neural networks (CNNs) comes from local structure modeling by convolutions and global representation learning via hierarchical or pyramid architectures. Convolution is defined as the integral of the product of two functions after one is reversed and shifted. The convolution of $f$ and $g$ is written as follows,

$$\text{Convolution:}\quad (f * g)(t) = \int_{\tau \in \Omega} g(\tau) f(t + \tau) d\tau, \tag{1}$$

where $\Omega$ is a neighborhood in a space. In deep learning, $f(t)$ usually is the feature at position $t$, *i.e.*, $f(t) = \boldsymbol{f}_t \in \mathbb{R}^{C \times 1}$, where $C$ is the number of input feature channels, and $g(\tau) \in \mathbb{R}^{C' \times C}$ is usually implemented as a parametric kernel function, where $C'$ is the number of output feature channels.

Traditionally, convolutions are mainly used for image processing (Krizhevsky et al., 2012) and natural language processing (Kim, 2014). Because positions of pixels and words or characters are regular, discrete convolutions are used to model their structures as follows,

$$\text{Discrete Convolution:}\quad \boldsymbol{f}_t' = \sum_{\Delta \in \mathcal{D}} g(\boldsymbol{\theta}_\Delta) \cdot \boldsymbol{f}_{t+\Delta}, \tag{2}$$

where $\cdot$ is matrix multiplication, and $\mathcal{D}$ is a regular and discrete neighborhood and can be seen as the kernel size of convolution. The key to discrete convolution is to employ an independent group of learnable weights for every displacement $\Delta$. As shown in Fig. 2(a), for a 2D discrete convolution with kernel size $3 \times 3$, there are nine independent groups of weights for the convolution: $\{\boldsymbol{\theta}_{(-1,-1)}, \boldsymbol{\theta}_{(-1,0)}, \boldsymbol{\theta}_{(-1,1)}, \boldsymbol{\theta}_{(0,-1)}, \boldsymbol{\theta}_{(0,0)}, \boldsymbol{\theta}_{(0,1)}, \boldsymbol{\theta}_{(1,-1)}, \boldsymbol{\theta}_{(1,0)}, \boldsymbol{\theta}_{(1,1)}\}$. For discrete convolution, $g$ is usually an identity function.

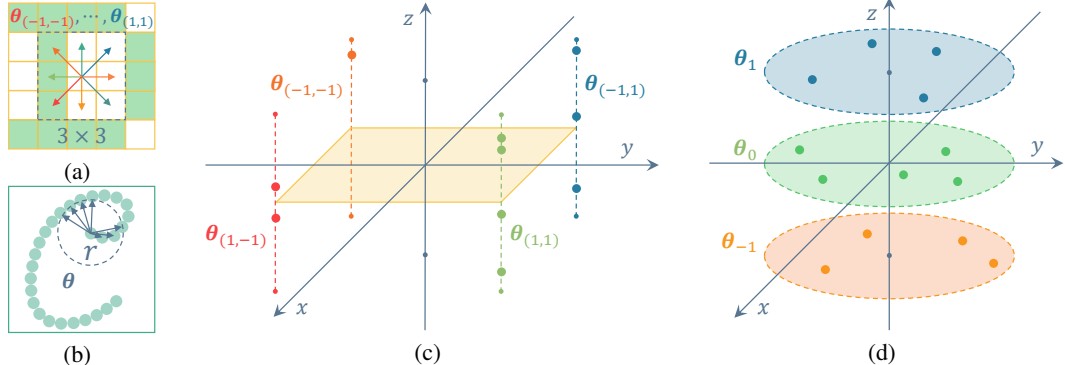

Figure 2: Illustration of discrete and continuous convolutions and examples of continuous-discrete spaces. **(a) 2D discrete convolution.** Pixels are distributed evenly in the 2D space and their coordinates are discrete. The convolution is capturing the structure of a $3 \times 3$ local area with nine independent groups of learnable weights. **(b) 2D continuous convolution.** For a continuous space, even in a small area, there can be countless types of displacements. In this case, the convolution is parameterized by the same weights $\boldsymbol{\theta}$ but directly takes the displacement as input. **(c) (1+2)D continuous-discrete space.** Points are distributed regularly on the 2D $xy$ plane but irregularly along the 1D $z$ axis. Four groups of weights are used to reflect the $2 \times 2$ discrete 2D positions. **(d) (2+1)D continuous-discrete space.** Points are located irregularly on the 2D $xy$ plane but regularly along the 1D $z$ axis. Three groups of weights are used to reflect the three discrete 1D positions.

Continuous convolutions are mainly used to handle irregular data, especially to point clouds. For an irregular and continuous space, even in an extremely small neighborhood $\mathcal{C}$, there can be countless displacements. Therefore, it is impossible to assign an independent group of learnable weights for every displacement $\delta$. In this case, the kernel function $g$ is parameterized by the same weights $\boldsymbol{\theta}$ for displacements but takes the displacement as an input,

$$\text{Continuous Convolution:} \quad \boldsymbol{f}'_t = \sum_{\delta \in \mathcal{C}} g(\delta; \boldsymbol{\theta}) \cdot \boldsymbol{f}_{t+\delta}. \tag{3}$$

We illustrate a continuous convolution in Fig. 2(b).

### 3.2 CONTINUOUS-DISCRETE CONVOLUTION

Besides exclusively regular or irregular, some data may simultaneously exhibit both regularity and irregularity. In this case, the neighborhood can be hybrid, *i.e.*, $\Omega = \mathcal{C} \times \mathcal{D}$, and each displacement in such a space consists of a discrete part $\Delta$ and a continuous part $\delta$. To model regular and irregular structures, we propose a continuous-discrete convolution (CDConv),

$$\text{Continuous} - \text{Discrete Convolution:} \quad \boldsymbol{f}'_t = \sum_{(\delta, \Delta) \in \mathcal{C} \times \mathcal{D}} g(\delta; \boldsymbol{\theta}_\Delta) \cdot \boldsymbol{f}_{t+\Delta+\delta}. \tag{4}$$

The crucial design of CDConv is to employ independent weights $\{\boldsymbol{\theta}_\Delta\}_{\Delta \in \mathcal{D}}$ for different regular displacements $\Delta$ but directly encode continuous displacements $\delta$ due to their irregularity. In this way, CDConv facilitates regular-irregular modeling and reduces the impact of the irregularity of continuous displacements on the regular and discrete structure modeling.

The proposed CDConv is a new class of convolution and we can derive different variants based on different combinations of continuous and discrete spaces. In this paper, we define $(i+j)$D as a hybrid space with an $i$D continuous structure and a $j$D discrete structure. The (1+2)D and (2+1)D continuous-discrete spaces are illustrated in Fig. 2(c) and Fig. 2(d), respectively.

### 3.3 (3+1)D CDCONV FOR PROTEIN MODELING

In a protein, the amino acids are linked by peptide bonds and form a chain. Let $(\boldsymbol{f}_1, \cdots, \boldsymbol{f}_N)$ denote the features of the amino acids in a protein, where $\boldsymbol{f}_t \in \mathbb{R}^{C \times 1}$ denotes the feature of the $t$-th amino

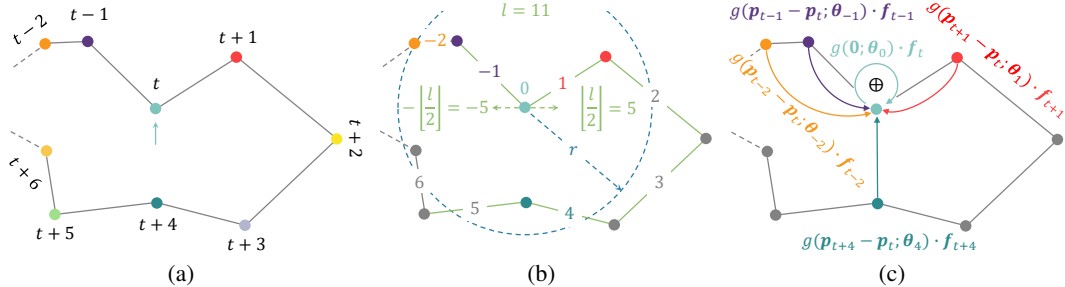

Figure 3: Illustration of the proposed (3+1)D CDConv for protein modeling. **(a)** The convolution is capturing the structure of a neighborhood centred at the $t$-the amino acid. **(b)** Based on the 1D sequential kernel size $l = 11$ and the 3D geometric radius $r$, the $(t-2)$-th, $(t-1)$-th, $t$-th, $(t+1)$-th and $(t+4)$-th amino acids are selected as neighbors and others are excluded because their sequential or geometric distances are too far away. **(c)** Based on the geometric displacement $(\boldsymbol{p}_{t+\Delta} - \boldsymbol{p}_t)$ and the sequential displacement $\Delta$, the kernel function $g(\boldsymbol{p}_{t+\Delta} - \boldsymbol{p}_t; \boldsymbol{\theta}_\Delta)$ generates the kernel weights for each selected neighbor. The neighbor features are transformed by the generated kernel weights with matrix multiplication, which are then aggregated as the new feature of the $t$-th amino acid. In this way, the local area structure of the $t$-th amino acid is captured.

acid and $N$ is the number of amino acids. In addition to the 1D order in the sequence, each amino acid has a 3D coordinate that specifies its spatial position, *i.e.*, $\boldsymbol{p}_t \in \mathbb{R}^{3 \times 1}$. The values of 1D sequential orders are regular and discrete while those of 3D coordinates are irregular and continuous non-integers. In this case, amino acids can be seen in a (3+1)D continuous-discrete space and we can employ the proposed convolution to capture the structure of a geometry-sequence neighborhood, resulting in the (3+1)D CDConv,

$$(3+1)\text{D} \quad \text{CDConv:} \quad \boldsymbol{f}'_t = \sum_{\|\boldsymbol{p}_{t+\Delta} - \boldsymbol{p}_t\| \leq r, \; -\lfloor l/2 \rfloor \leq \Delta \leq \lfloor l/2 \rfloor} g(\boldsymbol{p}_{t+\Delta} - \boldsymbol{p}_t; \boldsymbol{\theta}_\Delta) \boldsymbol{f}_{t+\Delta}, \quad (5)$$

where the neighborhood is defined with a 1D sequential kernel size $l$ and a 3D geometric radius $r$. We illustrate how (3+1)D CDConv captures the geometry-sequence structure of a local area in Fig. 3.

**Discussion**

*Comparison with existing methods.* The proposed (3+1)D CDConv is substantially different from existing methods that usually process geometric and sequential displacement together or in similar irregular manners. For example, GraphQA (Baldassarre et al., 2021) concatenates the 3D distance with a radial basis function and the 1D distance with the one-hot representation. The edge in GVP (Jing et al., 2021) embeds the unit vector in the 3D direction, the 3D distance with a radial basis function and the sinusoidal encoding of the 1D displacement. IEConv (Hermosilla & Ropinski, 2022) encodes normalized 1D displacements. GearNet (Zhang et al., 2022) directly concatenates 3D and 1D distances. Those methods can be summarised as $\gamma([\delta, \Delta])$ or $[\gamma(\delta), \eta(\Delta)]$, where $[\cdot, \cdot]$ demotes concatenation and $\gamma$ and $\eta$ are encoding functions. In contrast, (3+1)D CDConv does not encode or embed 1D displacements but employs independent weights for different discrete displacements. In this way, CDConv makes the most of the regularity of sequential orders. Moreover, because the units of length of the sequential and geometric spaces are not compatible, capturing the two structures in different ways improves the proposed CDConv for protein understanding.

*Comparison with two-stream 3D-CNN+1D-CNN architectures.* To make the most of the regularity of sequential orders and the irregularity of geometric coordinates, we can first employ an independent 3D continuous CNN branch and an independent 1D discrete CNN branch to extract the global geometric and sequential features, respectively, and then fuse the two features as the final protein representation (illustrated in Appendix D). However, because the two types of structures in proteins are modeled separately, the local (3+1)D structure is not properly captured. In contrast, CNNs based on our (3+1)D CDConv can simultaneously model the local geometry-sequence structure and thus extract more discriminative local features than 3D-CNN+1D-CNNs. As shown in Sec. 4.5.2, our (3+1)D CDConv-based CNN significantly outperforms the corresponding 3D-CNN+1D-CNN approach.

### 3.4 An Implementation for (3+1)D CDConv

There can be many ways to implement the kernel function $g$ for the proposed (3+1)D CDConv. In this paper, inspired by PointConv (Wu et al., 2019) and PSTNet (Fan et al., 2021), we implement $g$ as follows,

$$g(\boldsymbol{p}_{t+\Delta} - \boldsymbol{p}_t; \boldsymbol{\theta}_\Delta) = \boldsymbol{\theta}_\Delta \cdot (\boldsymbol{p}_{t+\Delta} - \boldsymbol{p}_t). \tag{6}$$

However, the above implementation is not rotationally invariant. Inspired by Hermosilla & Ropinski (2022), we replace the 3D displacement encoding with relative spatial encoding (Ingraham et al., 2019),

$$g(\boldsymbol{p}_{t+\Delta} - \boldsymbol{p}_t, \boldsymbol{O}_t, \boldsymbol{O}_{t+\Delta}; \boldsymbol{\theta}_\Delta) = \boldsymbol{\theta}_\Delta \cdot (\|\boldsymbol{p}_{t+\Delta} - \boldsymbol{p}_t\|, \boldsymbol{O}_t^T \cdot \frac{\boldsymbol{p}_{t+\Delta} - \boldsymbol{p}_t}{\|\boldsymbol{p}_{t+\Delta} - \boldsymbol{p}_t\|}, \boldsymbol{O}_t^T \cdot \boldsymbol{O}_{t+\Delta}), \tag{7}$$

where $\boldsymbol{O}_t = (\boldsymbol{b}_t, \boldsymbol{n}_t, \boldsymbol{b}_t \times \boldsymbol{n}_t)$ and $\boldsymbol{u}_t = \frac{\boldsymbol{p}_t - \boldsymbol{p}_{t-1}}{\|\boldsymbol{p}_t - \boldsymbol{p}_{t-1}\|}$, $\boldsymbol{b}_t = \frac{\boldsymbol{u}_t - \boldsymbol{u}_{t+1}}{\|\boldsymbol{u}_t - \boldsymbol{u}_{t+1}\|}$, $\boldsymbol{n}_t = \frac{\boldsymbol{u}_t \times \boldsymbol{u}_{t+1}}{\|\boldsymbol{u}_t \times \boldsymbol{u}_{t+1}\|}$ and $\boldsymbol{\theta}_\Delta \in \mathbb{R}^{C' \times 7}$. Finally, our (3+1)D CDConv is implemented as follows,

$$\boldsymbol{f}_t' = \sum_{\|\boldsymbol{p}_{t+\Delta} - \boldsymbol{p}_t\| \le r, \; -\lfloor l/2 \rfloor \le \Delta \le \lfloor l/2 \rfloor} g(\boldsymbol{p}_{t+\Delta} - \boldsymbol{p}_t, \boldsymbol{O}_t, \boldsymbol{O}_{t+\Delta}; \boldsymbol{\theta}_\Delta) \cdot \boldsymbol{f}_{t+\Delta}^T. \tag{8}$$

The matrix multiplication result is then reshaped from $\mathbb{R}^{C' \times C}$ to $\mathbb{R}^{C'C \times 1}$.

### 3.5 Continous-Discret Convolutional Neural Network

Based on the proposed CDConv, hierarchical deep neural networks can be built for global protein representation learning. To this end, we stack multiple CDConv layers and downsample amino acids as the number of layers increases.

To construct hierarchical networks, late layers should have larger receptive fields than early layers. For the 1D sequential kernel size $l$, we fix it because convolutions at late layers will capture relatively longer sequential regions due to amino acid downsampling. However, for the geometry modeling, we need to progressively enlarge the radius of each layer so that late layers have larger receptive fields.

Following Hermosilla et al. (2021) and Hermosilla & Ropinski (2022), we build a ResNet (He et al., 2016) with multiple ($h$) CDConv layers. After every two CDConv layers, a sequence sum pooling with downsampling rate 2 is performed to halve the protein resolution and build a pyramid architecture. For the $i$-th CDConv layer, the radius is set to $\lceil \frac{i}{2} + 1 \rceil r_o$ and the output dimension is set to $\lceil \frac{i}{2} \rceil c_o$, where $r_o$ and $c_o$ are the initial radius and the initial number of feature channels, respectively. In this way, the geometric receptive field and feature dimension progressively increase.

## 4 Experiments

### 4.1 Evaluation Tasks and Datasets

Following Hermosilla et al. (2021); Hermosilla & Ropinski (2022); Zhang et al. (2022), we evaluate the proposed method on four tasks: protein fold classification, enzyme reaction classification, gene ontology (GO) term prediction and enzyme commission (EC) number prediction. Protein fold classification includes three evaluation scenarios: fold, superfamily and family. GO term prediction includes three sub-tasks: biological process (BP), molecular function (MF) and cellular component (CC) ontology term prediction. More details about these tasks and datasets are shown in Appendix B.

Protein fold and enzyme reaction classification are single-label classification tasks. Mean accuracy is used as the evaluation metric. GO term and EC number prediction are multi-label classification tasks. The $F_{\max}$ accuracy is used as the evaluation metric (details are described in Appendix C).

### 4.2 Training Setup

We implement our method with the PyTorch-Geometric (PyG) library. The number of CDConv layers $h$ is set to 8. The sequential kernel size $l$ is set to 11 for fold classification, 25 for reaction

Table 1: Accuracy (%) of protein fold classification and enzyme catalytic reaction classification. [*]Results are from (Hermosilla et al., 2021). [†]Results are from (Zhang et al., 2022).

| Input | Method | Fold Classification | | | Enzyme |
| --- | --- | --- | --- | --- | --- |
| | | Fold | Superfamily | Family | Reaction |
| 1D | CNN (Shanehsazzadeh et al., 2020)[†] | 11.3 | 13.4 | 53.4 | 51.7 |
| | ResNet (Rao et al., 2019)[†] | 10.1 | 7.21 | 23.5 | 24.1 |
| | LSTM (Rao et al., 2019)[†] | 6.41 | 4.33 | 18.1 | 11.0 |
| | Transformer (Rao et al., 2019)[†] | 9.22 | 8.81 | 40.4 | 26.6 |
| 3D | GCN (Kipf & Welling, 2017)[*] | 16.8 | 21.3 | 82.8 | 67.3 |
| | GAT (Velickovic et al., 2018)[†] | 12.4 | 16.5 | 72.7 | 55.6 |
| | 3D CNN (Derevyanko et al., 2018)[*] | 31.6 | 45.4 | 92.5 | 72.2 |
| 3D+Topo | IEConv (atom level) (Hermosilla et al., 2021) | 45.0 | 69.7 | 98.9 | 87.2 |
| (3+1)D | GraphQA (Baldassarre et al., 2021)[*] | 23.7 | 32.5 | 84.4 | 60.8 |
| | GVP (Jing et al., 2021)[†] | 16.0 | 22.5 | 83.8 | 65.5 |
| | IEConv (residue level) (Hermosilla & Ropinski, 2022) | 47.6 | 70.2 | 99.2 | 87.2 |
| | GearNet (Zhang et al., 2022) | 28.4 | 42.6 | 95.3 | 79.4 |
| | GearNet-IEConv (Zhang et al., 2022) | 42.3 | 64.1 | 99.1 | 83.7 |
| | GearNet-Edge (Zhang et al., 2022) | 44.0 | 66.7 | 99.1 | 86.6 |
| | GearNet-Edge-IEConv (Zhang et al., 2022) | 48.3 | 70.3 | 99.5 | 85.3 |
| | CDConv (ours) | **56.7** | **77.7** | **99.6** | **88.5** |

classification and 15 for GO term and EC number prediction. The initial radius $r_o$ is set to 4. More details of implementation and training setup are provided in Appendix E [1].

## 4.3 COMPARISON WITH STATE-OF-THE-ART

We compare our method with existing 1D-only, 3D-only and (3+1)D methods. Results are shown in Table 1 and Table 2. The IEConv method includes an atom-level variant (Hermosilla et al., 2021) and an amino-acid-level variant (Hermosilla & Ropinski, 2022). Because the atom-level IEConv is based on the 3D coordinates of atoms and the topology structure of bonds between atoms, we denote the input of atom-level IEConv as "3D+Topo".

Note that none of the existing methods achieved state-of-the-art accuracy on all of the four tasks. In contrast, our method significantly outperforms all the existing methods. For example, on the superfamily fold classification, our CDConv outperforms the previous state-of-the-art method, *i.e.*, GearNet-Edge-IEConv, by 6.9%. On the BP ontology term prediction, our CDConv outperforms the previous best method, *i.e.*, the residue-level IEConv, by 3.2%. This demonstrates the effectiveness of the proposed CDConv that employs discrete and continuous approaches for regular sequence and irregular geometry modeling, respectively.

## 4.4 HOW DOES CONTINUOUS-DISCRETE CNN CAPTURE PROTEIN STRUCTURE?

To investigate the behavior of our CNN, we visualize the outputs of CDConvs. We treat the norm $\|f\|$ of the output features as the activation strength of amino acids and visualize the activation strength in Fig. 4. Visualizations of proteins of different sizes for all CDConv layers are illustrated in Appendix T. Recall that, to build a pyramid architecture, the CNN downsamples amino acids after every two layers. The early layers (*e.g.*, CDConv 2 and CDConv 4) tend to focus on the central region of the protein. Then, the activation spreads to the entire protein so that the late layers (*e.g.*, CDConv 6 and CDConv 8) can capture the whole structure. This indicates that the central structure may provide the most informative clues in early recognition and may be more important than other areas.

To verify the explanation for the above phenomenon, we mask the same number (20%) of amino acids in the central and surface areas, respectively. As shown in Fig. 5(a), masking the central area leads to more accuracy drops, indicating that amino acids in central regions may be more important than those in surface areas. More details and results are shown in Appendix J.

---

[1]https://github.com/hehefan/Continuous-Discrete-Convolution

Table 2: $F_{max}$ of gene ontology term prediction and enzyme commission number prediction. [*]Results are from (Wang et al., 2021). [†]Results are from (Zhang et al., 2022).

| Input | Method | Gene Ontology | | | Enzyme |
|---|---|---|---|---|---|
| | | BP | MF | CC | Commission |
| 1D | CNN (Shanehsazzadeh et al., 2020)[†] | 0.244 | 0.354 | 0.287 | 0.545 |
| | ResNet (Rao et al., 2019)[†] | 0.280 | 0.405 | 0.304 | 0.605 |
| | LSTM (Rao et al., 2019)[†] | 0.225 | 0.321 | 0.283 | 0.425 |
| | Transformer (Rao et al., 2019)[†] | 0.264 | 0.211 | 0.405 | 0.238 |
| 3D | GCN (Kipf & Welling, 2017)[†] | 0.252 | 0.195 | 0.329 | 0.320 |
| | GAT (Velickovic et al., 2018) | 0.284[*] | 0.317[*] | 0.385[*] | 0.368[†] |
| | 3D CNN (Derevyanko et al., 2018)[†] | 0.240 | 0.147 | 0.305 | 0.077 |
| (3+1)D | GraphQA (Baldassarre et al., 2021)[†] | 0.308 | 0.329 | 0.413 | 0.509 |
| | GVP (Jing et al., 2021) | 0.326[*] | 0.426[*] | 0.420[*] | 0.489[†] |
| | IEConv (residue level) (Hermosilla & Ropinski, 2022) | 0.421 | 0.624 | 0.431 | - |
| | GearNet (Zhang et al., 2022) | 0.356 | 0.503 | 0.414 | 0.730 |
| | GearNet-IEConv (Zhang et al., 2022) | 0.381 | 0.563 | 0.422 | 0.800 |
| | GearNet-Edge (Zhang et al., 2022) | 0.403 | 0.580 | 0.450 | 0.810 |
| | GearNet-Edge-IEConv (Zhang et al., 2022) | 0.400 | 0.581 | 0.430 | 0.810 |
| | CDConv (ours) | **0.453** | **0.654** | **0.479** | **0.820** |

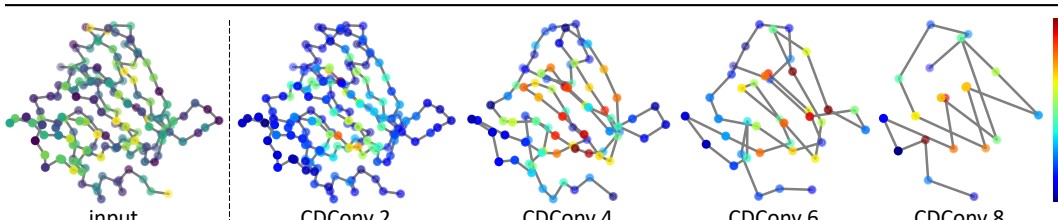

Figure 4: Visualization of how the proposed network captures protein structure. The color of input dots indicates amino acid types and that of CDConv outputs denotes the strength of the activation. The early layers tend to focus on the central region and the late layers try to capture the whole structure. This may indicate that the central region is more important than other areas to proteins.

## 4.5 ABLATION STUDY

### 4.5.1 IMPACT OF CONTINUOUS GEOMETRY AND DISCRETE SEQUENCE MODELING

In this paper, we propose to use continuous and discrete approaches to model the geometry and sequence structures, respectively. To verify the effectiveness, we conduct cross-experiments. To model the regular sequence structure in a continuous manner, we directly encode the sequential displacement $\Delta$. To capture the irregular geometry structure in a discrete manner, 3D coordinates are voxelized with size 0.5. Results are shown in Table 3.

First, when replacing discrete modeling with directly encoding on the sequence structure capture, the accuracy drops. This verifies our motivation that regular structure should be modeled in a regular manner. Second, when using the discrete method to modeling the geometry structure, the accuracy drops significantly. This is because quantization errors are inevitable during voxelization and voxelized coordinates cannot precisely describe the geometry structure.

### 4.5.2 COMPARISON WITH TWO-STREAM 3D-CNN+1D-CNN

To avoid the impact of irregular and regular modeling on each other, one can employ an independent continuous 3D-CNN and an independent discrete 1D-CNN to build a two-stream network, resulting in 3D-CNN+1D-CNN. We compare our CDConv-based CNN with the corresponding 3D-CNN+1D-CNN. The 3D-CNN is implemented by PointConv with relative spatial encoding. Each of the 3D-CNN stream and the 1D-CNN stream consist of the same number of layers as our CDConv-based CNN's. As shown in Table 3, our CDConv-based CNN surpasses the 3D-CNN+1D-CNN by a large margin. Moreover, compared to the 3D geometry-only continuous CNN, the two-stream method

Table 3: Influence of continuous and discrete modeling on 3D and 1D structure capture, respectively, and comparison with two-stream 3D-CNN+1D-CNN. "RI" indicates rotational invariance.

| 3D Geometry | 1D Sequence | RI | Fold Classification | | | Enzyme | Gene Ontology | | | Enzyme |
|---|---|---|---|---|---|---|---|---|---|---|
| Structure | Structure | | Fold | Superfamily | Family | Reaction | BP | MF | CC | Commission |
| Discrete | ✗ | ✗ | 31.5 | 42.6 | 85.9 | 72.6 | 0.336 | 0.526 | 0.355 | 0.664 |
| Continuous | ✗ | ✓ | 41.9 | 65.5 | 99.0 | 86.5 | 0.426 | 0.605 | 0.435 | 0.737 |
| ✗ | Discrete | - | 13.9 | 20.2 | 86.6 | 71.1 | 0.371 | 0.548 | 0.436 | 0.561 |
| ✗ | Continuous | - | 9.1 | 12.8 | 71.1 | 64.0 | 0.348 | 0.488 | 0.417 | 0.482 |
| Discrete | Discrete | ✗ | 38.9 | 48.2 | 90.6 | 77.3 | 0.378 | 0.560 | 0.410 | 0.701 |
| Continuous | Continuous | ✓ | 51.7 | 74.2 | 99.2 | 87.1 | 0.432 | 0.618 | 0.446 | 0.782 |
| Discrete | Continuous | ✗ | 34.2 | 43.8 | 87.0 | 74.1 | 0.340 | 0.544 | 0.369 | 0.681 |
| Two-Stream 3D-CNN+1D-CNN | | ✓ | 43.5 | 65.4 | 98.4 | 86.1 | 0.412 | 0.601 | 0.414 | 0.758 |
| Continuous | Discrete | ✓ | **56.7** | **77.7** | **99.6** | **88.5** | **0.453** | **0.654** | **0.479** | **0.820** |

(a) impact of surface/central mask  (b) geometric radius $r_o$  (c) sequential kernel size $l$

Figure 5: **(a)** To investigate which area is more important, we mask the same number of amino acids in central and surface regions and evaluate the impact on protein recognition. **(b)** Impact of initial geometric radius. **(c)** Impact of sequential kernel size on protein modeling.

does not significantly improve the accuracy or even leads to worse accuracy. These indicate that the 3D-CNN+1D-CNN strategy is not effective to capture the local (3+1)D structure of proteins and the global representations of the two structures may not benefit each other.

### 4.5.3 IMPACT OF GEOMETRIC RADIUS AND SEQUENTIAL KERNEL SIZE

The geometric radius and sequential kernel size control the range of the spatial and chain structure to be modeled, respectively. As shown in Fig. 5(b) and Fig. 5(c), using too small radius and kernel size cannot capture sufficient structure information and using too large radius and kernel size will decrease the discriminativeness of spatial local structure for modeling, thus leading to inferior accuracy. The impact of the number of channels and CDConv layers are shown in Appendix F and in Appendix G.

## 5 CONCLUSION

We first propose a new class of convolutions, *i.e.*, CDConv, which uses continuous and discrete approaches to model the irregular and regular structures in data, respectively. Then, we implement a (3+1)D CDConv to model the sequence and geometry structures in proteins. We apply CDConv to a range of protein modeling tasks and achieve the new state-of-the-art accuracy. Moreover, by investigating the behavior of CDConv, we find that amino acids in central regions may be more important than those in surface areas to proteins, which can be further verified via experimental efforts. As a new class of convolutions, the applications of CDConv are not limited to protein modeling and can be other scenarios where data exhibits both regularity and irregularity. The proposed (3+1)D CDConv can also be used to recognize non-coding RNAs and may be potentially improved with different implementations. However, as a convolution, our CDConv requires a predefined and fixed geometric radius and sequential kernel size to capture local structure, which is somewhat rigid. This can be alleviated in future studies by integrating self-attention to adaptively capture the local structure.

ACKNOWLEDGMENTS

This research is supported by the Agency for Science, Technology and Research (A*STAR) under its AME Programmatic Funding Scheme (#A18A2b0046).

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

## A    PROBLEM FORMULATION

To evaluate the proposed CDConv, we apply it to the protein classification problem. Classification is a fundamental task for pattern recognition and representation learning. Specifically, given a protein, which consists of a list of amino acid coordinates $\boldsymbol{P} \in \mathbb{R}^{3 \times N}$ and a list of amino acid types $\boldsymbol{F} \in \mathbb{R}^{1 \times N}$, where $N$ is the number of amino acids, the goal of protein classification is to map $\boldsymbol{P}$ and $\boldsymbol{F}$ to a prediction vector $\boldsymbol{y} \in \mathbb{R}^k$, where $k$ is the number of protein classes or categories,

$$\boldsymbol{y} = \Phi(\boldsymbol{P}, \boldsymbol{F}; \Theta).$$

The $\Phi$ is a deep neural network and $\Theta$ is its parameters.

Then, for single-label classification, the softmax function is usually used to convert $\boldsymbol{y}$ to prediction probabilities, *i.e.*, $\mathrm{softmax}(\boldsymbol{y})$. The class with the highest probability will be considered as the final prediction.

For multi-label classification, the sigmoid function is usually used to convert $\boldsymbol{y}$ to binary prediction probabilities, *i.e.*, $\sigma(\boldsymbol{y})$. Those classes whose probabilities are higher than a predefined threshold, *e.g.*, 0.5, will be considered as the final predictions.

## B    DETAILS OF EVALUATION TASKS AND DATASETS

**Protein Fold Classification.** Protein fold classification is important in the study of the relationship between protein structure and protein evolution. The fold classes indicate protein secondary structure compositions, orientations and connection orders. We follow Hermosilla et al. (2021) to conduct protein fold classification on the training/validation/test splits of the SCOPe 1.75 data set of Hou et al. (2018), which in total contains 16,712 proteins with 1,195 fold classes. The 3D coordinates of the proteins were collected from the SCOPe 1.75 database (Murzin et al., 1995). The data set provides three different evaluation scenarios. 1) Fold, in which proteins from the same superfamily are not used during training. 2) Superfamily, in which proteins from the same family are not provided during training. 3) Family, in which proteins of the same family are available during training. Mean accuracy is used as the evaluation metric.

**Enzyme Reaction Classification.** Enzyme reaction classification can be seen as a protein function classification task, which is based on the enzyme-catalyzed reaction according to all four levels of the Enzyme Commission (EC) number (Webb, 1992). We use the dataset collected by Hermosilla et al. (2021), which includes 384 four-level EC classes and 29,215/2,562/5,651 proteins for training/validation/test, respectively. Mean accuracy is used as the evaluation metric.

**Gene Ontology Term Prediction.** This task aims to predict the functions of a protein via multiple Gene Ontology (GO) terms, which can be seen as a multi-label classification task. Following Gligorijević et al. (2021), we classify proteins into hierarchically related functional classes organized into three ontologies: biological process (BP) with 1,943 classes, molecular function (MF) with 489 classes and cellular component (CC) with 320 classes. The dataset contains 29,898/3,322/3,415 proteins for training/validation/test, respectively. The $F_{\max}$ accuracy is used as the evaluation metric.

**Enzyme Commission Number Prediction.** Different from enzyme reaction classification, this task aims to predict the three-level and four-level 538 EC numbers. We use the training/validation/test splits of Gligorijević et al. (2021), which contains 15,550/1,729/1,919 proteins in total. EC number prediction is also a multi-label classification task. We use $F_{\max}$ as the evaluation metric.

For GO term and EC number prediction, we follow the multi-cutoff splits in (Gligorijević et al., 2021) to ensure that the test set only contains PDB chains with sequence identity no more than 95% to the training set, which is also used in (Wang et al., 2021; Zhang et al., 2022). Accuracy with all cutoffs is shown in Appendix K.

## C    EVALUATION METRIC $F_{\max}$

Multi-label classification can be seen as consisting of a series of binary classification tasks. The protein-centric maximum F-Score, *i.e.*, $F_{\max}$, as defined in Gligorijević et al. (2021), can be used to measure the accuracy of multi-label classification. Suppose $\lambda \in [0, 1]$ is a decision threshold, $p_i^j$

is the prediction probability for the $j$-th class of the $i$-th protein, $b_i^j \in \{0, 1\}$ is the corresponding binary class label and $J$ is the number of classes. F-Score is based on the precision and recall of the predictions for each protein,

$$\text{precision}_i(\lambda) = \frac{\sum_j^J \left( (p_i^j \geq \lambda) \cap b_i^j \right)}{\sum_j^J (p_i^j \geq \lambda)}, \quad \text{recall}_i(\lambda) = \frac{\sum_j^J (p_i^j \geq \lambda)}{\sum_j^J b_i^j},$$

and the average precision and recall over all proteins are defined as follows,

$$\text{precision}(\lambda) = \frac{\sum_i^N \text{precision}_i(\lambda)}{\sum_i^N \left( \left( \sum_j^J (p_i^j \geq \lambda) \right) \geq 1 \right)}, \quad \text{recall}(\lambda) = \frac{\sum_i^N \text{recall}_i(\lambda)}{N},$$

where $N$ is the number of proteins. Then, $\text{F}_{\max}$ is the maximum F-Score among the different thresholds $\lambda$ tested in the range $[0, 1]$,

$$\text{F}_{\max} = \max_{\lambda \in [0,1]} \left\{ \frac{2 \times \text{precision}(\lambda) \times \text{recall}(\lambda)}{\text{precision}(\lambda) + \text{recall}(\lambda)} \right\}.$$

## D  FRAMEWORK ILLUSTRATION

The power of CNNs comes from local structure modeling by convolutions and global representation learning via hierarchical architectures. We illustrate our CDConv-based CNN in Fig. 6(a). Our CDConv captures the (3+1)D structure in each local area. To build the pyramid framework, amino acids are downsampled along CDConv layers. The final (3+1)D feature is used for classification.

To avoid the impact of the regularity of sequential orders and the irregularity of geometric coordinates on each other, we can employ a two-stream 3D-CNN+1D-CNN framework, as illustrated in Fig. 6(b). First, an independent 3D continuous CNN branch and an independent 1D discrete CNN branch are used to extract the global geometric and sequential features, respectively. Then, the two features are fused as the final protein representation. However, because the two types of structures in proteins are modeled separately, the local (3+1)D structure is not properly captured. In contrast, our (3+1)D CDConv-based CNNs can simultaneously model the local geometry-sequence structure and thus extract more discriminative local features than 3D-CNN+1D-CNN architectures.

## E  MORE DETAILS OF IMPLEMENTATION AND TRAINING SETUP

We provide more details of implementation and training setup in Table 4. Our model is implemented based on PyTorch-Geometric 2.0.4 and PyTorch 1.10.0 with CUDA 11.3.1 and cuDNN 8.2.0. Experiments are conducted on a single Nvidia Quadro RTX A5000 GPU.

Table 4: More details of implementation and training setup.

| Hyper-parameter | Fold | Enzyme Reaction | GO-BP | GO-MF | GO-CC | EC |
|---|---|---|---|---|---|---|
| Batch size | 8 | 8 | 24 | 24 | 64 | 24 |
| # Channels ($c_o$) | 256 | 256 | 256 | 128 | 128 | 256 |

In data pre-processing, amino acid coordinates are normalized as follows,

$$\bar{p} = \frac{1}{N} \sum_{i=1}^N p_i, \quad p_i = p_i - \bar{p},$$

where $p_i$ is the coordinate of $i$-th amino acid and $N$ is the number of amino acids. During training, we add the simple isotropic additive gaussian noise (Jitter) to amino acid coordinates for data augmentation.

In practice, we find that directly neglecting the neighboring amino acids whose sequential distances are greater than $\lfloor l/2 \rfloor$ but geometric distances are less than $r$ is not conducive to optimization. In the

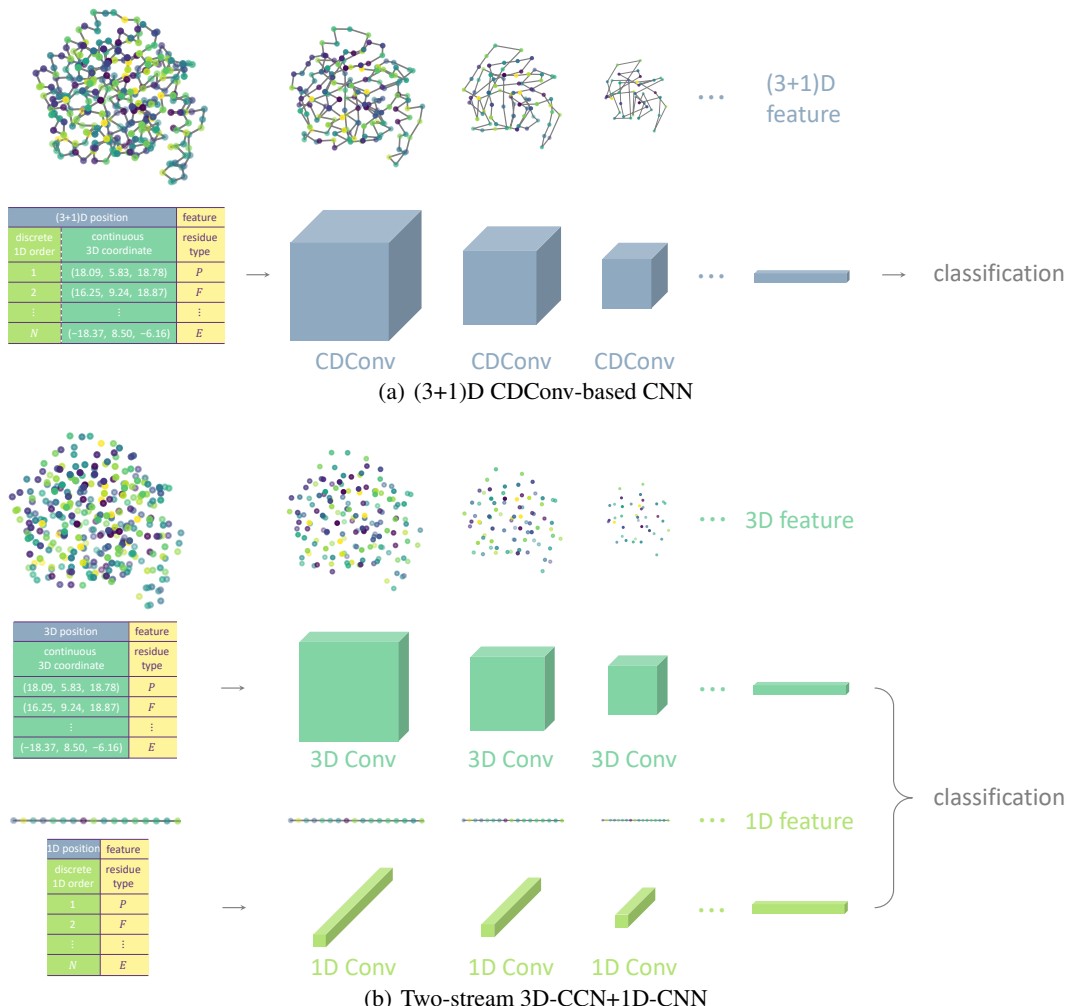

Figure 6: Illustration of (3+1)D CDConv-based CNN and two-stream 3D-CCN+1D-CNN.

implementation, we include those geometric but non-sequential neighbors as follows,

$$
\boldsymbol{f}'_t = \sum_{\|\boldsymbol{p}_{t+\Delta}-\boldsymbol{p}_t\|\leq r,\ -\lfloor l/2\rfloor\leq\Delta\leq\lfloor l/2\rfloor} g(\boldsymbol{p}_{t+\Delta}-\boldsymbol{p}_t,\boldsymbol{O}_t,\boldsymbol{O}_{t+\Delta};\boldsymbol{\theta}_\Delta)\cdot\boldsymbol{f}_{t+\Delta}^T \quad +
$$

$$
\sum_{\|\boldsymbol{p}_{t+\Delta}-\boldsymbol{p}_t\|\leq r,\ \Delta<-\lfloor l/2\rfloor} g(\boldsymbol{p}_{t+\Delta}-\boldsymbol{p}_t,\boldsymbol{O}_t,\boldsymbol{O}_{t+\Delta};\boldsymbol{\theta}_-)\cdot\boldsymbol{f}_{t+\Delta}^T \quad +
$$

$$
\sum_{\|\boldsymbol{p}_{t+\Delta}-\boldsymbol{p}_t\|\leq r,\ \Delta>\lfloor l/2\rfloor} g(\boldsymbol{p}_{t+\Delta}-\boldsymbol{p}_t,\boldsymbol{O}_t,\boldsymbol{O}_{t+\Delta};\boldsymbol{\theta}_+)\cdot\boldsymbol{f}_{t+\Delta}^T,
$$

where $\boldsymbol{\theta}_-$ and $\boldsymbol{\theta}_+$ are two additional groups of learnable weights for those "special" neighbors. In this way, networks are not significantly sensitive to the sequential kernel size hyper-parameter $l$, as shown in Fig. 5(c).

## F  IMPACT OF THE NUMBER OF FEATURE CHANNELS

Similar to most CNNs, we use multiple CDConv layers to build hierarchical or pyramid frameworks for protein representation learning. For the $i$-th CDConv layer, the number of channels is set to $\lceil \frac{i}{2}\rceil c_o$, where $c_o$ is the initial number of channels. In this way, the feature dimension progressively increases when CNNs become deep.

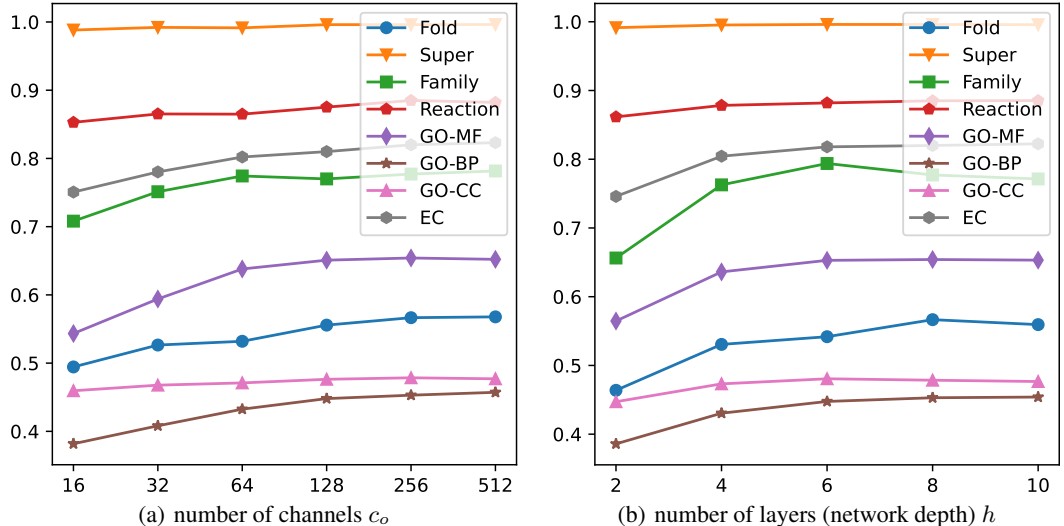

Figure 7: Impact of the number of channels $c_o$ and the number of CDConv layers ($h$) on protein recognition. Usually, $c_o \geq 128$ and $h \geq 6$ are sufficient to achieve satisfactory accuracy.

As shown in Fig. 7(a), using too few channels cannot carry enough information and therefore lead to inferior accuracy. However, when employing too many channels, the accuracy does not significantly increase. This means that the number of weights or parameters is enough. To further improve the accuracy, we may need to develop more effective CDConv implementations or network architectures.

## G   IMPACT OF THE NUMBER OF LAYERS (NETWORK DEPTH)

By default, we use eight CDConv layers ($h = 8$) to build our CNN for protein representation learning. In this section, we investigate the influence of the number of layers. As shown in Fig. 7(b), when $h$ is too small, networks fail to effectively capture the protein structure. However, when networks are too deep, the accuracy even slightly decreases. This is because, with depth increasing, the receptive fields of late layers are too large, which decreases the discriminativeness of local structures. The other reason is that too many layers may lead to overfitting.

Note that for protein fold classification and GO-CC term prediction, using six CDConv layers ($h = 6$) already achieves satisfactory accuracy. However, for the consistency in visualization, we still use eight CDConv layers.

## H   IMPACT OF GEOMETRY AND SEQUENCE MODELING ON PROTEIN UNDERSTANDING

The proposed CDConv employs the 3D geometry structure and the 1D sequence structure in proteins for recognition. In this section, we analyze the influence of each structure.

As shown in Table 3, both the two structures contribute to protein understanding. For example, without the sequence structure modeling, the fold classification accuracy decreases from 56.7% to 41.9%. Without the geometry structure modeling, the GO-BP term prediction $F_{max}$ decreases from 0.453 to 0.371.

For the two types of modeling, geometry structure contributes more than sequence structure. For example, the 3D-only CDConv achieves 65.5% on superfamily fold classification while the 1D-only CDConv achieves 20.8%. This indicates that geometry structure is more discriminative than sequence structure for protein recognition.

## I  IMPACT OF RELATIVE SPATIAL ENCODING

Table 5: Comparison of relative spatial encoding and direct 3D displacement encoding for protein fold classification (%), enzyme reaction classification (%), gene ontology term prediction ($F_{max}$) and enzyme commission prediction ($F_{max}$).

| Method | Fold Classification | | | Enzyme | Gene Ontology | | | Enzyme |
| --- | --- | --- | --- | --- | --- | --- | --- | --- |
| | Fold | Superfamily | Family | Reaction | BP | MF | CC | Commission |
| 3D displacement encoding | 43.3 | 57.6 | 95.6 | 86.0 | 0.349 | 0.594 | 0.405 | 0.757 |
| Relative spatial encoding | 56.7 | 77.7 | 99.6 | 88.5 | 0.453 | 0.654 | 0.479 | 0.820 |

To make our method rotationally invariant, we replace the 3D displacement encoding in Point-Conv (Wu et al., 2019) with relative spatial encoding (Ingraham et al., 2019). The details are as follows,

$$\text{3D displacement encoding:} \quad \boldsymbol{\theta}_\Delta \cdot (\boldsymbol{p}_{t+\Delta} - \boldsymbol{p}_t),$$

$$\text{relative spatial encoding:} \quad \boldsymbol{\theta}_\Delta \cdot \left(\|\boldsymbol{p}_{t+\Delta} - \boldsymbol{p}_t\|, \boldsymbol{O}_t^T \cdot \frac{\boldsymbol{p}_{t+\Delta} - \boldsymbol{p}_t}{\|\boldsymbol{p}_{t+\Delta} - \boldsymbol{p}_t\|}, \boldsymbol{O}_t^T \cdot \boldsymbol{O}_{t+\Delta}\right).$$

In this section, we conduct experiments to evaluate the effectiveness of relative spatial encoding. As shown in Table 5, relative spatial encoding significantly improves the accuracy, demonstrating the effectiveness of rotational invariance for protein modeling.

## J  MORE DETAILS AND RESULTS OF THE STUDY ON THE IMPORTANCE OF SURFACE AND CENTRAL AMINO ACIDS

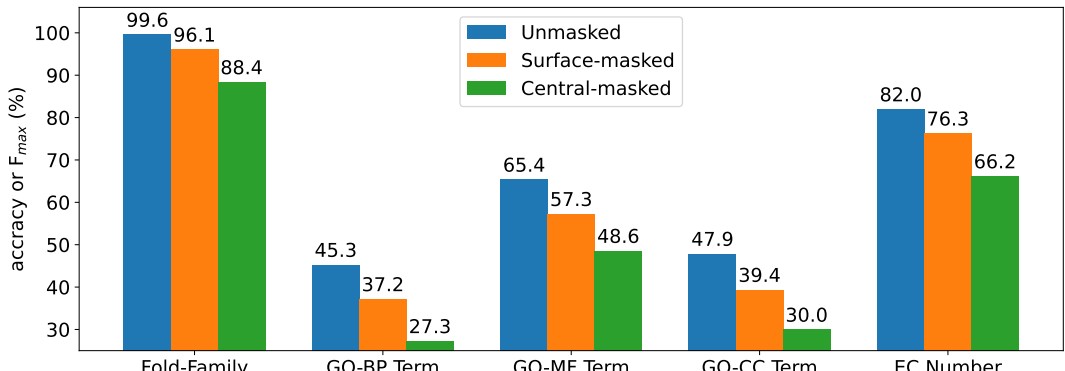

Figure 8: Impact of surface and central mask on protein recognition. Masking central area leads to more drops of accuracy. This may indicate that amino acids in central regions may be more important than those in surface areas.

To investigate which of the surface and central regions is more important, we mask the same number (20%) of amino acids in the surface and central areas, respectively. Specifically, we first calculate the center of a protein as follows,

$$\bar{\boldsymbol{p}} = \frac{1}{N} \sum_{i=1}^{N} \boldsymbol{p}_i,$$

where $N$ is the number of amino acids and $\boldsymbol{p}_i$ denotes the 3D coordinate of the $i$-th amino acid. For surface region masking, we remove the 20% farthest amino acids from the center based on the geometry distance. For central region masking, we remove the 20% closest amino acids to the center.

The results of family fold classification, GO term prediction and EC number prediction are shown in Fig. 8. Note that amino acids in central areas are usually denser than those in surface regions. This

means that, when masking the same number of amino acids, we actually mask more surface regions. Even so, masking central area leads to more drops of accuracy. This indicates that amino acids in central regions may be more important than those in surface areas. However, future experimental efforts are required to further verify this phenomenon.

## K GO TERM AND EC NUMBER PREDICTION UNDER DIFFERENT SEQUENCE CUTOFFS

Table 6: $F_{max}$ on GO term and EC number prediction under different cutoffs. Our method is compared with CNN (Shanehsazzadeh et al., 2020), ResNet (Rao et al., 2019), LSTM (Rao et al., 2019), Transformer (Rao et al., 2019), GCN (Kipf & Welling, 2017), GearNet and GearNet-Edge (Zhang et al., 2022). [*]Results are from (Zhang et al., 2022).

| Cutoff | 30% | 40% | 50% | 70% | 95% | 30% | 40% | 50% | 70% | 95% |
|---|---|---|---|---|---|---|---|---|---|---|
| Method | GO-BP | | | | | GO-MF | | | | |
| CNN[*] | 0.197 | 0.195 | 0.197 | 0.211 | 0.244 | 0.238 | 0.243 | 0.256 | 0.292 | 0.354 |
| ResNet[*] | 0.230 | 0.230 | 0.234 | 0.249 | 0.280 | 0.282 | 0.288 | 0.308 | 0.347 | 0.405 |
| LSTM[*] | 0.194 | 0.192 | 0.195 | 0.205 | 0.225 | 0.223 | 0.229 | 0.245 | 0.276 | 0.321 |
| Transformer[*] | 0.267 | 0.265 | 0.262 | 0.262 | 0.264 | 0.184 | 0.187 | 0.195 | 0.204 | 0.211 |
| GCN[*] | 0.251 | 0.250 | 0.248 | 0.248 | 0.252 | 0.180 | 0.183 | 0.187 | 0.194 | 0.195 |
| GearNet | 0.309 | 0.309 | 0.315 | 0.336 | 0.356 | 0.382 | 0.397 | 0.425 | 0.474 | 0.503 |
| GearNet-Edge | 0.345 | 0.347 | 0.354 | 0.378 | 0.403 | 0.444 | 0.461 | 0.490 | 0.537 | 0.580 |
| CDConv (ours) | **0.381** | **0.390** | **0.401** | **0.428** | **0.453** | **0.533** | **0.553** | **0.577** | **0.621** | **0.654** |
| Method | GO-CC | | | | | EC | | | | |
| CNN[*] | 0.258 | 0.257 | 0.260 | 0.263 | 0.387 | 0.366 | 0.361 | 0.372 | 0.429 | 0.545 |
| ResNet[*] | 0.277 | 0.273 | 0.280 | 0.278 | 0.304 | 0.409 | 0.412 | 0.450 | 0.526 | 0.605 |
| LSTM[*] | 0.263 | 0.264 | 0.269 | 0.270 | 0.283 | 0.247 | 0.249 | 0.270 | 0.333 | 0.425 |
| Transformer[*] | 0.378 | 0.382 | 0.388 | 0.395 | 0.405 | 0.167 | 0.173 | 0.175 | 0.197 | 0.238 |
| GCN[*] | 0.318 | 0.318 | 0.320 | 0.323 | 0.329 | 0.245 | 0.246 | 0.246 | 0.280 | 0.320 |
| GearNet | 0.381 | 0.385 | 0.393 | 0.398 | 0.414 | 0.557 | 0.570 | 0.615 | 0.693 | 0.730 |
| GearNet-Edge | 0.394 | 0.394 | 0.401 | 0.408 | 0.450 | 0.625 | 0.646 | 0.694 | 0.757 | 0.810 |
| CDConv (ours) | **0.428** | **0.435** | **0.440** | **0.451** | **0.479** | **0.634** | **0.659** | **0.702** | **0.768** | **0.820** |

For GO term and EC number prediction, we can employ different cutoff splits (Gligorijević et al., 2021): the proteins in the test set are split to have < 30%, < 40%, < 50%, < 70 % and < 95 % similarity to the training set, respectively. The results are shown in Table 6. Our CDConv significantly outperforms the existing methods in most experimental scenarios. For example, for GO-MF term prediction under cutoff 40%, our CDConv outperforms GearNet-Edge by 9.2%. For GO-CC term prediction under cutoff 70%, our CDConv outperforms GearNet-Edge by 4.3%. This demonstrates the effectiveness of the proposed method.

## L COMPARISON WITH PRETRAINING METHODS

Recently, pretraining or self-supervised learning on 3D protein structures has gained a lot of attention in the fields of protein modeling and structural bioinformatics. The goal of those methods is to exploit unlabeled protein data to boost accuracy by designing self-supervised or contrastive objective functions. For example, Hermosilla & Ropinski (2022) proposed to train networks by maximizing the similarity between representations from sub-structures of the same protein, and minimizing the similarity between sub-structures from different proteins. Zhang et al. (2022) proposed to maximize the similarity between representations of different augmented views of the same protein while minimizing the agreement between views of different proteins, and to predict amino acid types, the Euclidean distance between nodes connected in the protein graph, the angle between nodes and the dihedral angles between edges. However, as mentioned by Fan et al. (2022), it is unclear whether one or a few self-supervision tasks are sufficient for learning effective representations and which

Table 7: Accuracy (%) of protein fold classification. [*]Results are from (Hermosilla & Ropinski, 2022). [†]Results are from (Zhang et al., 2022).

| Method | Pretraining Dataset (Size) | Fold Classification | | |
|---|---|---|---|---|
| | | Fold | Superfamily | Family |
| DeepFRI[*] | Pfam (10M) | 15.3 | 20.6 | 73.2 |
| ESM-1b[†] | UniRef50 (24M) | 26.8 | 60.1 | 97.8 |
| ProtBERT-BFD[*] | BFD (2.1B) | 26.6 | 55.8 | 97.6 |
| IEConv (amino level) | PDB (476K) | 50.3 | **80.6** | 99.7 |
| GearNet-Edge-IEConv with Multiview Contrast | AlphaFoldDB (805K) | 54.1 | 80.5 | **99.9** |
| GearNet-Edge-IEConv with Residue Type Prediction | AlphaFoldDB (805K) | 48.8 | 71.0 | 99.4 |
| GearNet-Edge-IEConv with Distance Prediction | AlphaFoldDB (805K) | 50.9 | 73.5 | 99.4 |
| GearNet-Edge-IEConv with Angle Prediction | AlphaFoldDB (805K) | 56.5 | 76.3 | 99.6 |
| GearNet-Edge-IEConv with Dihedral Prediction | AlphaFoldDB (805K) | 51.8 | 77.8 | 99.6 |
| IEConv (residue level) | - | 47.6 | 70.2 | 99.2 |
| GearNet-Edge-IEConv | - | 48.3 | 70.3 | 99.5 |
| CDConv (ours) | - | **56.7** | 77.7 | 99.6 |

Table 8: Accuracy (%) of enzyme catalytic reaction classification and $F_{max}$ of gene ontology term prediction and enzyme commission number prediction. [+]Results are from (Wang et al., 2021). [*]Results are from (Hermosilla & Ropinski, 2022). [†]Results are from (Zhang et al., 2022).

| Method | Pretraining Dataset (Size) | Enzyme Reaction | Gene Ontology | | | Enzyme Commission |
|---|---|---|---|---|---|---|
| | | | BP | MF | CC | |
| DeepFRI | Pfam (10M) | 63.3[*] | 0.399[†] | 0.465[†] | 0.460[†] | 0.631[†] |
| ESM-1b[†] | UniRef50 (24M) | 83.1 | 0.470 | 0.657 | 0.488 | 0.864 |
| ProtBERT-BFD | BFD (2.1B) | 72.2[*] | 0.279[+] | 0.456[+] | 0.408[+] | 0.838[†] |
| LM-GVP | UniRef100 (216M) | - | 0.417 | 0.545 | **0.527** | 0.664[†] |
| IEConv (amino level) | PDB (476K) | 88.1 | 0.468 | **0.661** | 0.516 | - |
| GearNet-Edge with Multiview Contrast | AlphaFoldDB (805K) | 87.5 | **0.490** | 0.654 | 0.488 | **0.874** |
| GearNet-Edge with Residue Type Prediction | AlphaFoldDB (805K) | 86.6 | 0.430 | 0.604 | 0.465 | 0.843 |
| GearNet-Edge with Distance Prediction | AlphaFoldDB (805K) | 87.5 | 0.448 | 0.616 | 0.464 | 0.839 |
| GearNet-Edge with Angle Prediction | AlphaFoldDB (805K) | 86.8 | 0.458 | 0.625 | 0.473 | 0.853 |
| GearNet-Edge with Dihedral Prediction | AlphaFoldDB (805K) | 87.0 | 0.458 | 0.626 | 0.465 | 0.859 |
| IEConv (residue level) | - | 87.2 | 0.421 | 0.624 | 0.431 | - |
| GearNet-Edge | - | 86.6 | 0.403 | 0.580 | 0.450 | 0.810 |
| CDConv (ours) | - | **88.5** | 0.453 | 0.654 | 0.479 | 0.820 |

task would be considered more. Doersch & Zisserman (2017) even attempted to combine several self-supervised tasks to obtain better visual representations.

Different from those pretraining or self-supervised learning works, our method focuses on designing an effective fundamental operation to capture the protein structure, instead of task-related objective or loss functions. An effective fundamental operation is general and independent of specific training strategies and may improve the baseline for the pretraining or self-supervised learning works.

In this section, to show the effectiveness, we compare our CDConv to pretraining or self-supervised learning methods: DeepFRI (Gligorijević et al., 2021), DeepFRI (Gligorijević et al., 2021), ESM-1b (Rives et al., 2021), ProtBERT-BFD (Elnaggar et al., 2021), LM-GVP (Wang et al., 2021), amino-acid-level IEConv (Hermosilla & Ropinski, 2022) and GearNet-based methods (Zhang et al., 2022).

As shown in Table 7 and Table 8, without any pretraining or self-supervised learning, our CDConv achieves comparable accuracy with those methods and even outperforms them on fold and enzyme reaction classification.

Table 9: Accuracy (%) of distance-only-based "completely-rotationally-invariant" (3+1)D CDConv on protein fold classification. [*]Results are from (Hermosilla et al., 2021). [†]Results are from (Zhang et al., 2022).

| Method | Fold | Superfamily | Family |
|---|---|---|---|
| GraphQA (Baldassarre et al., 2021)[*] | 23.7 | 32.5 | 84.4 |
| GVP (Jing et al., 2021)[†] | 16.0 | 22.5 | 83.8 |
| IEConv (residue level) (Hermosilla & Ropinski, 2022) | 47.6 | 70.2 | 99.2 |
| GearNet (Zhang et al., 2022) | 28.4 | 42.6 | 95.3 |
| GearNet-IEConv (Zhang et al., 2022) | 42.3 | 64.1 | 99.1 |
| GearNet-Edge (Zhang et al., 2022) | 44.0 | 66.7 | 99.1 |
| GearNet-Edge-IEConv (Zhang et al., 2022) | 48.3 | 70.3 | 99.5 |
| Distance-only-based CDConv w/o distance normalization | 46.6 | 73.2 | 99.3 |
| Distance-only-based CDConv w/ distance normalization | 48.6 | 73.5 | 99.4 |
| Relative-spatial-encoding-based CDConv | **56.7** | **77.7** | **99.6** |

## M  ROTATION EQUIVARIANCE FOR OTHER SCENARIOS

In this paper, we propose a new class of convolution, i.e., CDConv, to make the most of the dual discrete and continuous nature of the data to avoid the impact of regular and irregular modeling on each other. The key design of CDConv is to employ independent learnable weights for different regular displacements but directly encode irregular displacements. As a prototype, CDConv can be instantiated to various instances. In this paper, we implement a (3+1)D CDConv for protein representation learning. When implementing the 3D geometry modeling part, we use the relative spatial encoding technique.

The relative spatial encoding is independent of our CDConv and can be replaced with other techniques. When for other scenarios, we can integrate the corresponding rotation equivariance technique into CDConv. Moreover, there can be a low-bound solution to completely avoid the rotational variance problem: only use the distance information and do not model direction,

$$g\big(\boldsymbol{p}_{t+\Delta} - \boldsymbol{p}_t; \boldsymbol{\theta}_\Delta\big) = \boldsymbol{\theta}_\Delta \cdot (\|\boldsymbol{p}_{t+\Delta} - \boldsymbol{p}_t\|/r),$$

where the distance is normalized with the neighbor searching radius $r$. Finally, the new (3+1)D CDConv is implemented as follows,

$$\boldsymbol{f}'_t = \sum_{\|\boldsymbol{p}_{t+\Delta} - \boldsymbol{p}_t\| \leq r,\ -\lfloor l/2 \rfloor \leq \Delta \leq \lfloor l/2 \rfloor} g(\boldsymbol{p}_{t+\Delta} - \boldsymbol{p}_t; \boldsymbol{\theta}_\Delta) \cdot \boldsymbol{f}^T_{t+\Delta}.$$

Then, we evaluate this distance-only-based "completely-rotationally-invariant" (3+1)D CDConv on protein fold classification. As shown in Table 9, the new (3+1)D CDConv still achieves comparable accuracy with the previous state-of-the-art methods.

## N  COMPUTATIONAL EFFICIENCY AND SCALABILITY

To investigate the performance of our method on relatively large proteins, we collect the proteins that contain more than 500 amino acids from the enzyme reaction classification dataset, leading to a subset with large proteins, as shown in Table 10.

However, because the training data significantly reduces (from 29,215 to 2,498 proteins), the subset of large proteins may lead to inferior accuracy. Therefore, for a fair comparison, we randomly sample a normal subset that contains the same number of proteins as that of large proteins.

Then, we conduct enzyme reaction classification on the two subsets with a single Nvidia Quadro RTX A5000 GPU and Intel(R) Xeon(R) Silver 4214R CPU @ 2.40GHz. We also report the mean running time per prediction. As shown in Table 11, our CDConv can still effectively and efficiently recognize enzyme reactions.

Table 10: Details of the original dataset, the subset of large proteins and the normal subset of randomly sampled proteins for enzyme reaction classification.

| Dataset | Training | | | | Valiadation | | | | Test | | | |
|---|---|---|---|---|---|---|---|---|---|---|---|---|
| | Largest | Smallest | Average | Size | Largest | Smallest | Average | Size | Largest | Smallest | Average | Size |
| Original | 3,725 | 18 | 296.2 | 29,215 | 1,314 | 24 | 346.0 | 2,562 | 1,119 | 20 | 293.1 | 5,651 |
| Large | 3,725 | 500 | 722.3 | 2,498 | 1,314 | 500 | 711.6 | 319 | 1,119 | 500 | 611.0 | 450 |
| Normal | 2,650 | 23 | 293.6 | 2,498 | 1,314 | 31 | 363.1 | 319 | 1,119 | 25 | 299.3 | 450 |

Table 11: Performance comparison of CDConv on enzyme reaction classification between large and normal proteins.

| Subset | Accuracy | Running Time |
|---|---|---|
| Normal Proteins | 67.7% | 10.2 ms |
| Large Proteins | 65.3% | 13.2 ms |

## O   DISCUSSION ON ATTENTION-BASED METHODS

Recently, attention-based methods (Jumper et al., 2021; Baek et al., 2021; Humphreys et al., 2021; Fuchs et al., 2020) are also widely used for protein-related problems, such as protein structure prediction or generation. As we discussed in Sec. 5, attention-based techniques may potentially improve our method.

Using deep learning to model a kind of data can be divided into two parts: 1) searching related elements (*e.g.*, amino acids) and 2) encoding the local structure of the related elements. Attention-based methods can adaptively reflect the correlation degree of two elements and therefore are good at searching related elements. Our CDConv focuses on encoding the region of the related elements by making the most of the dual discrete and continuous nature of the data. The two kinds of methods may facilitate each other, which can be studied in the future.

## P   INFLUENCE OF MISSING COORDINATES

When some residues' coordinates are not available, we can set their coordinates to a pre-defined value, *e.g.*, $(0, 0, 0)$. In this way, our CDConv can still be used for protein modeling.

To evaluate the effectiveness of this strategy, we randomly select a certain percentage of amino acids and set their coordinates to $(0, 0, 0)$ in training and test and conduct protein fold classification. As shown in Table 12, even with 20% missing coordinates, our CDConv still achieves promising results.

## Q   INFLUENCE OF VOXEL SIZE FOR DISCRETIZING 3D COORDINATES

In our experiments, when using discrete convolutions for 3D geometry structure modeling, we follow the practice in most discrete CNNs and employ kernel size 3, which means there are $3 \times 3 \times 3 = 27$ group of learnable weights for each convolution.

Because the computation resource is limited, we fix the kernel size to $3 \times 3 \times 3$. In this case, there is a trade-off, which is also a limitation, to use discrete convolutions for 3D geometry modeling:

- To increase the receptive field size, networks have to reduce the resolution of proteins by increasing voxel size.
- When using high-resolution proteins, convolutions can only capture a small region each time.

In summary, due to the limited computation resource, we have to fix the kernel size and set it to a small range. In this way, if we want to increase the receptive field, we can accordingly reduce the resolution. If we want a higher resolution, the receptive field has to be relatively "shrunk".

Table 12: Accuracy (%) of CDConv with missing coordinates on protein fold classification.

| Percentage of Residues with Missing Coordinates | Fold | Superfamily | Family |
|---|---|---|---|
| 0% | **56.7** | **77.7** | **99.6** |
| 5% | 54.6 | 75.8 | 99.5 |
| 10% | 51.6 | 74.0 | 99.5 |
| 20% | 49.8 | 73.2 | 99.3 |
| 30% | 44.0 | 69.1 | 99.1 |
| 40% | 42.8 | 65.0 | 98.8 |

Table 13: Influence of voxel size for discrete convolution on protein fold classification.

| Voxel Size | Fold | Superfamily | Family |
|---|---|---|---|
| 0.5 | 31.5 | 42.6 | 85.9 |
| 1.0 | 30.6 | 32.3 | 73.4 |
| 2.0 | 26.7 | 27.8 | 66.4 |
| 4.0 | 19.5 | 22.9 | 51.5 |
| 8.0 | 15.0 | 13.2 | 35.0 |

In experiments, we found setting the voxel size for discretizing 3D coordinates to 0.5 achieves the best accuracy. The results for other voxel sizes are shown in Table 13. When increasing the voxel size, although the $3 \times 3 \times 3$ can capture a relatively large receptive field, the accuracy does not increase. This is because increasing voxel size results in low resolution, which cannot properly describe protein structures.

## R EMPLOYING PROTEIN STRUCTURE PREDICTION MODELS FOR PROTEIN CLASSIFICATION

In recent years, deep-learning-based methods have made tremendous achievements in protein structure prediction, such as AlphaFold (Jumper et al., 2021) and RoseTTAFold (Baek et al., 2021). In this section, we investigate using protein structure prediction methods for protein classification.

There is a difference in output between protein structure prediction and classification.

- **Protein structure prediction**. The goal of structure prediction is to make residue-wise predictions. The output is $\mathbb{R}^{N \times C}$ or $\mathbb{R}^{N \times C \times C}$, where $N$ is the number of amino acids and $C$ is the number of prediction channels. For structure prediction, $C = 3$.

- **Protein classification**. Classification aims to make global predictions, in which a global protein representation vector is learned, *i.e.*, $\mathbb{R}^C$. In our method, $C = 1024$ or $C = 2048$. Because CNNs are usually hierarchical and pyramid, which captures structures from local to global, it is natural for CNNs to learn global representations.

Another difference is that protein structure prediction usually exploits external knowledge, *e.g.*, Multiple Sequence Alignments (MSAs) and protein templates. For classification, existing methods are usually based on the given limited training data.

In this section, we convert RoseTTAFold (Baek et al., 2021) for protein classification. Specifically, we use the outputs of the feature extractor in RoseTTAFold: MSA feature $\in \mathbb{R}^{N \times 384}$, pair representation $\in \mathbb{R}^{N \times N \times 288}$ and xyz $\in \mathbb{R}^{N \times 3 \times 3}$. To generate a global representation vector based on these outputs, we first perform sum on the second dimension of the pair representation and get $\mathbb{R}^{N \times 288}$ and flatten xyz as $\mathbb{R}^{N \times 9}$. Then, we concatenate MSA feature, processed pair representation and xyz: $\mathbb{R}^{N \times (384+288+9)} = \mathbb{R}^{N \times 681}$. Third, we append a fully-connected layer with ReLU to increase the dimension of the concatenation from 681 to 1024, *i.e.*, $\mathbb{R}^{N \times 1024}$. Last, we sum those features and generate a global representation $\mathbb{R}^{1024}$. In this way, classification can be performed.

We use the modified RoseTTAFold for protein fold classification. The results are shown in Table 14. When MSAs are not exploited, we follow Baek & Baker (2022) to only use the peptide chain itself.

Table 14: Accuracy (%) of the modified RoseTTAFold by us on protein fold classification. *Results are from (Hermosilla et al., 2021). †Results are from (Zhang et al., 2022).

| Method | Fold | Superfamily | Family |
|---|---|---|---|
| GraphQA (Baldassarre et al., 2021)* | 23.7 | 32.5 | 84.4 |
| GVP (Jing et al., 2021)† | 16.0 | 22.5 | 83.8 |
| IEConv (residue level) (Hermosilla & Ropinski, 2022) | 47.6 | 70.2 | 99.2 |
| GearNet (Zhang et al., 2022) | 28.4 | 42.6 | 95.3 |
| GearNet-IEConv (Zhang et al., 2022) | 42.3 | 64.1 | 99.1 |
| GearNet-Edge (Zhang et al., 2022) | 44.0 | 66.7 | 99.1 |
| GearNet-Edge-IEConv (Zhang et al., 2022) | 48.3 | 70.3 | 99.5 |
| CDConv (ours) | **56.7** | **77.7** | **99.6** |
| RoseTTAFold | 22.7 | 28.2 | 85.5 |
| RoseTTAFold (with MSA) | 24.2 | 31.4 | 87.0 |
| RoseTTAFold (with Template) | 31.8 | 39.5 | 91.1 |
| RoseTTAFold (with MSA and Template) | 33.4 | 40.4 | 92.3 |

Table 15: Accuracy (%) of generalized CDConv with missing coordinates on protein fold classification.

| Percentage of Residues with Missing Coordinates | Fold | Superfamily | Family |
|---|---|---|---|
| 0% | **56.7** | **77.7** | **99.6** |
| 5% | 54.8 | 76.2 | 99.6 |
| 10% | 52.9 | 75.7 | 99.5 |
| 20% | 50.8 | 74.5 | 99.5 |
| 30% | 47.1 | 73.1 | 99.2 |
| 40% | 46.0 | 69.2 | 99.0 |

When protein templates are not exploited, we use the input protein geometry structure as the only template. Probably because the architecture of RoseTTAFold is not suitable for global representation learning, it achieves limited accuracy. Converting RoseTTAFold to a pyramid architecture may boost the accuracy for protein classification, which can be studied in the future.

We notice that MSAs and protein templates help classification. They may play a data augmentation role in protein classification. Moreover, protein templates help more than MSAs. This may indicate that the 3D geometry structure is more important than the 1D sequence structure for protein recognition.

## S  GENERALIZED CDCONV FOR EXPLICITLY DEALING WITH MISSING 3D COORDINATES

To explicitly deal with missing 3D coordinates, we generalize our CDConv as follows,

$$\boldsymbol{f}'_t = \sum_{\|\boldsymbol{p}_{t+\Delta} - \boldsymbol{p}_t\| \leq r, \ -\lfloor l/2 \rfloor \leq \Delta \leq \lfloor l/2 \rfloor} \mathbf{1}_{\text{exsiting}} \times g_1(\boldsymbol{p}_{t+\Delta} - \boldsymbol{p}_t, \boldsymbol{O}_t, \boldsymbol{O}_{t+\Delta}; \boldsymbol{\theta}^1_\Delta) \cdot \boldsymbol{f}^T_{t+\Delta} + \mathbf{1}_{\text{missing}} \times g_2(\boldsymbol{\theta}^2_\Delta) \cdot \boldsymbol{f}^T_{t+\Delta},$$

where $\mathbf{1}_{\text{exsiting}}$ indicates that the neighbor's coordinate exists and $\mathbf{1}_{\text{missing}}$ indicates that the coordinate is missing. When coordinate is available, CDConv models the geometry-sequence structure with the function $g_1(\cdot; \boldsymbol{\theta}^1_\Delta)$. When coordinate is not available, CDConv only models the sequence structure with the function $g_2(\boldsymbol{\theta}^2_\Delta)$.

When all coordinates are missing, the generalized CDConv becomes 1DConv. Therefore, 1DConv can be seen as a special case of the generalized CDConv.

We employ the generalized CDConv for protein fold classification. As shown in Table 15, the extended CDConv can effectively alleviate the coordinate-missing problem.

## T  VISUALIZATION OF CDCONV OUTPUTS

We visualize the outputs of all the CDConv layers in our CNNs in Fig. 9. Recall that the CNN downsamples amino acids after every two layers. No matter for large or small proteins, the early layers mainly focus on the central regions of the proteins. Then, the activation spreads to the entire proteins to capture the whole structure. This indicates that the central region of proteins may be more important than other areas. However, experimental efforts are required to further verify this phenomenon.

## U  INVESTIGATE WHETHER THE CENTRAL REGION OF PROTEIN IS REALLY MORE IMPORTANT

In this paper, we find that amino acids in central regions may be more important than those in surface areas via those visualizations of CDConv outputs. To further verify this phenomenon, we mask the same number of amino acids in the central and surface areas, respectively. Then, we find that masking the central area leads to more accuracy drops, indicating that amino acids in central regions may be more important than those in surface areas.

Even so, this phenomenon may be an artefact of the proposed method. To verify the phenomenon, biochemical experiments are required. However, we currently do not have such resources to conduct biochemical experiments. We hope that the protein engineering community would verify the phenomenon in the future.

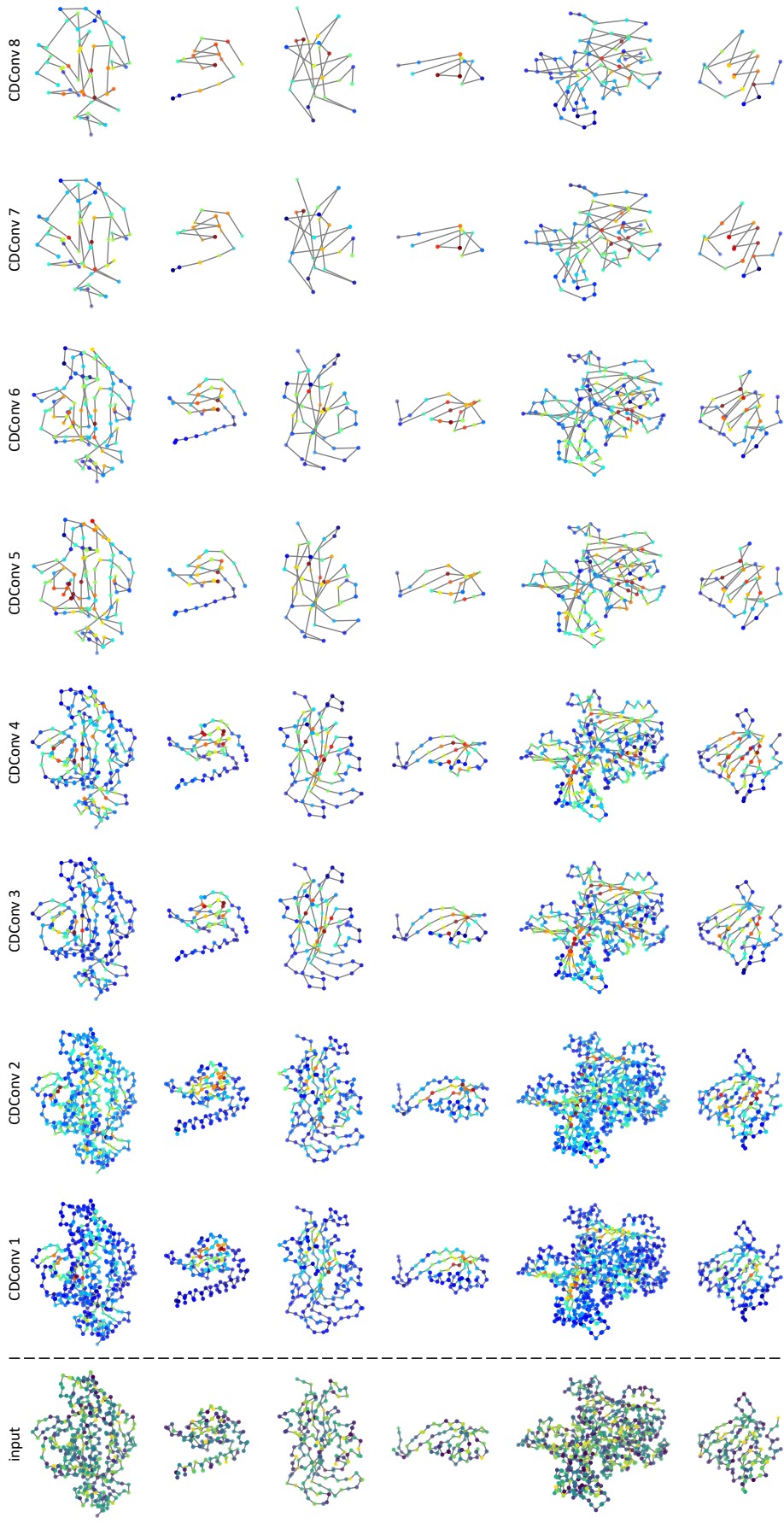

Figure 9: Visualization of how the proposed network captures protein structure. The color of input dots indicates amino acid type and the color of CDConv outputs denotes the strength of the activation. The early layers tend to focus on the central region and the late layers tend to capture the whole structure. This may indicate that the central region is more important than other areas for protein recognition.

