# OpenReview forum: "Continuous-Discrete Convolution for Geometry-Sequence Modeling in Proteins"
_ICLR.cc/2023/Conference — ICLR 2023 poster_

### Official Review · Reviewer_JFLd · 2022-10-24

**Confidence:** 4
**Correctness:** 3
**Technical Novelty And Significance:** 3
**Empirical Novelty And Significance:** 3
**Recommendation:** 6

**Clarity, Quality, Novelty And Reproducibility:**

Clarity: Well written, easy to follow.

Quality: Good.

Novelty: The motivation is well founded, and the proposed method is novel and well captures the dual discrete and continuous nature of protein structures.

Reproducibility: Authors have claimed that the code will be released.

**Strength And Weaknesses:**

Pros:
1. The observation of regular sequence structure and irregular geometry structure is valuable, which leads to the main motivation of this paper. Previous works indeed fail to perfectly cover both aspects of protein structures.
2. The proposed relative spatial encoding is rotational invariant, which is critical for efficiency model optimization. In addition, such rotational-invariant implementation is more computationally efficient than spherical harmonics based ones, e.g., SE(3)-Transformer.

Cons:
1. I am not sure whether the proposed method can properly deal with protein structure where some residues have missing 3D coordinates, possibly due to the difficulty in experimental determination. According to Equation (7), the construction of relative spatial encoding relies on 3D coordinates of adjacent residues’ C-alpha atoms.
2. As described in Section 4.2, the sequential kernel size is set to different values in different tasks (fold classification: 11; reaction classification: 25; Go term and EC number prediction: 15). Does this indicate that the proposed model is somehow sensitive to the choice of sequential kernel size? How should this be alleviated?
3. In Table 3, discrete convolution for 3D geometry structures is evaluated. For discretized 3D coordinates, it would be hard to preserve the rotational equivariance/invariance, thus explicit data augmentation should be adopted for more sufficient training. Could you please provide additional details on how data augmentation is adopted here?
4. In Table 3, it would be nice to explicitly highlight which methods are rotational invariant (thus does not require data augmentation) and which are not, so as for easier comparison.
5. In Section 4.5.1, the voxel size for discretizing 3D coordinates is set to 0.5A. Since the geometry radius is 4A, this means that the discrete convolutional kernel is of size 8 x 8 x 8, which may have the risk of over-parameterization and under-fitting. Will this affect the performance of such baseline methods?

**Summary Of The Paper:**

In this paper, authors propose continuous-discrete convolution (CDConv) for joint modeling of geometric and sequential protein structures. This is motivated by the observation of regular sequence structure and irregular geometry structure of proteins. The proposed instantiation of CDConv is rotational invariant, which does not require additional data augmentation and is efficient to optimize. Empirical evaluation on several benchmark datasets demonstrates the effectiveness of the proposed method.

**Summary Of The Review:**

The motivation of joint modeling of regular sequence structure and irregular geometry structure of proteins is reasonable. The proposed CDConv achieves this goal by decomposing convolutional kernels into independent weights for different sequential displacements and continuous relative spatial encoding, which is rotational invariant and efficient to optimize. However, some comparison with alternative convolutional mechanisms is not well conducted, as the difference in rotation invariance and necessity for data augmentation may also affect these results. Additional discussion may be needed.

---

> ### Author Response · Authors · 2022-11-13
> **Part I: Experiments with Missing Coordinates, Sensitivity to Sequential Kernel Size**
>
> We thank you for acknowledging our motivation **"is well founded"** and **"the proposed method is novel"**. We also thank you for your detailed comments.
>
> ### 1. Influence of missing coordinates
>
> When some residues' coordinates are not available, we can set their coordinates to a pre-defined value, e.g., $(0,0,0)$. In this way, our CDConv can still be used for protein modeling.
>
> To evaluate the effectiveness of this strategy, we randomly select a certain percentage of amino acids and set their coordinates to $(0,0,0)$ in training and test and conduct protein fold classification. As shown in Table 1, even with 20% missing coordinates, our CDConv still achieves promising results.
>
> **Table 1:** Accuracy (%) of CDConv with missing coordinates on protein fold classification.
>
> | Percentage of Residues with Missing Coordinates | Fold | Superfamily | Family |
> | ---- | ---- | ---- | ----- |
> | 0% | 56.7 | 77.7 | 99.6 |
> | 5% | 54.6 |  75.8 | 99.5 |
> | 10% | 51.6 | 74.0 | 99.5 |
> | 20% |49.8 | 73.2 | 99.3 |
> | 30% | 44.0 | 69.1 | 99.1 |
> | 40% | 42.8 | 65.0 | 98.8 |
>
> **We added “Appendix P: Influence of Missing Coordinates”  in the revision.**
>
>
> ### 2. Sensitivity to sequential kernel size
>
> As presented in “Appendix E: More Details of Implementation and Training Setup”, in the implementation, we include those geometric but non-sequential neighbors with another two groups of independent parameters, i.e., $\boldsymbol\theta_{-}$  ( for sequential displacement $< -l/2$ ) and $\boldsymbol\theta_{+}$  ( for sequential displacement $>l/2$ ), instead of directly neglecting them.  In this way, networks are not significantly sensitive to the sequential kernel size $l$, as shown in Fig. 5(c) in the paper.  To make it clear, we fix the sequential kernel size to 15. As shown in Table 2, CDConv is not significantly sensitive to the sequential kernel size. Moreover, the results are still the-state-of-the-art.
>
>
> **Table 2:** Influence of sequential kernel size to CDConv on different protein recognition tasks. Note that $l=15$ is the optimal value for BP, MF, CC and Enzyme Commission.
> | Sequential Kernel Size | Fold | Superfamily | Family | Enzyme Reaction | BP | MF | CC | Enzyme Commission |
> | --- | --- | --- | --- | --- | --- | --- | --- | --- |
> | Optimal $l$ per Task | 56.7 | 77.7 | 99.6 | 88.5 | 0.453 | 0.654 | 0.479 | 0.820 |
> | $l$=15 | 53.3 | 76.6 | 99.5 | 87.8 | 0.453 | 0.654 | 0.479 | 0.820 |
>
>
> ### 3. Data augmentation for discrete convolution on 3D geometry modeling
>
> When using discrete convolutions for 3D structure modeling, we first perform a rotation data augmentation on coordinates and then voxelize them to regular data.  The rotation angles of each axis are randomly sampled from $[-0.2\pi, 0.2\pi]$.
>
> ### 4. Rotational invariance information in Table 3
>
> Thank you for the valuable suggestion. We added the rotational invariance information in Table 3. Please see the revision. In summary, when using discrete convolutions for 3D geometry structure modeling, these methods are not rotationally invariant.

---

> > ### Author Response · Authors · 2022-11-15
> > **Part II: Details of Discrete 3D Convolution for Geometry Modeling**
> >
> > ### 5. Influence of voxel size for discretizing 3D coordinates
> >
> > In our experiments, when using discrete convolutions for 3D geometry structure modeling, we follow the practice in most discrete CNNs and employ kernel size 3, which means there are $3 \times 3 \times 3 = 27$ groups of learnable parameters for each convolution.
> >
> > Because the computation resource is limited, we have to fix kernel size. If we want to increase the receptive field with the same kernel size, we can increase voxel size. In this way, the convolution can capture the structure of a relatively large area.   If we want to reduce the receptive field with the fixed kernel size,  we can reduce the voxel size. Then,  the same convolution can capture the structure of a relatively small area each time.  Actually, there is a trade-off, which is also a limitation, to use discrete convolutions for 3D geometry modeling:
> >
> > ```
> > Based on the same kernel size,
> >
> > a) To increase the receptive field size, networks have to reduce the resolution of proteins by increasing voxel size.
> >
> > b) When using high-resolution proteins, convolutions can only capture a small region each time.
> > ```
> >
> >
> > In summary, due to the limited computation resource, we have to fix the kernel size and set it to a small range. In this way, if we want to increase the receptive field, we can accordingly reduce the resolution. If we want a higher resolution, the receptive field has to be relatively “shrunk”.
> >
> > In experiments, we found setting the voxel size for discretizing 3D coordinates to 0.5 achieves the best accuracy. The results for other voxel sizes are shown in Table 3.
> >
> > **Table 3:** Influence of voxel size for discrete convolution on protein fold classification.
> > | Voxel Size | Fold | Superfamily | Family |
> > | ---- | ---- | ---- | ---- |
> > | 0.5 | 31.5 | 42.6 | 85.9 |
> > | 1.0 | 30.6 | 32.3 | 73.4 |
> > | 2.0 | 26.7 | 27.8 | 66.4 |
> > | 4.0 | 19.5 | 22.9 | 51.5 |
> > | 8.0 | 15.0 | 13.2 | 35.0 |
> >
> > When increasing the voxel size, although the $3 \times 3 \times 3$ can capture a relatively large receptive field, the accuracy does not increase.
> > This is because increasing voxel size results in low resolution, which cannot properly describe protein structures.
> >
> > Dear reviewer, we hope our efforts could address your concerns. We would appreciate it if you could kindly consider raising your rating. Your support is significantly important to our paper. Should you need further information, please let us know. We look forward to hearing from you soon. Thank you!

---

### Official Review · Reviewer_hLLm · 2022-10-24

**Confidence:** 4
**Clarity, Quality, Novelty And Reproducibility:** The paper is clearly written and orig…
**Correctness:** 4
**Technical Novelty And Significance:** 3
**Empirical Novelty And Significance:** 3
**Recommendation:** 6

**Strength And Weaknesses:**

Strengths:
* The idea is a simple and practically effective melding of two widely used ideas in protein modeling with a large practical and empirical upside.
* The empirical performance of the method is a strength.  It has state of the art performance on all classification tasks benchmarked.  Can the authors comment on situations in which they expect the approach might not perform as well?



Weaknesses:
* The authors should comment on attention-based alternative methods.  In particular the approach of constructing an NxN dimensional pair representation influenced by both 1D and 3D information, and using it to computed biases on attention weights is at this point very well recognized to be extremely effective (at least in the context of protein structure prediction).  Presumably this same mechanism /architecture could be applied to these classification tasks.  Is there a reason why the authors do not compare to this?
* I find the terminology “(3+1)D” to be very off-putting and distracting since my mind collapses (3+1) into (4), which I suppose is not intended.


**Summary Of The Paper:**

Discrete convolutions have been widely used on 2D images and 1D protein sequences.  And continuous convolutions have been widely used on 3D data such as point clouds and to some extent on 3D protein structures.  The paper proposes to apply both at once and shows one can obtain improved empirical results with such an approach on protein classification tasks.

However, attention based mechanisms e.g. as in AlphaFold and RosettaFold have been largely replacing convolutions over the past 2 years.  The authors should comment on these methods as an alternative.  E.g. why not repurpose the AlphaFold (or a relevant subset of its architecture) to combine the 1D and 3D information?

**Summary Of The Review:**

I recommend accepting the paper because it clearly describes a simple and effective new architecture for protein classification problems.

---

> ### Author Response · Authors · 2022-11-13
> **Discussion on Attention-based Methods**
>
> We thank you for acknowledging that our paper is **"original"** and **"practically effective"**. We also thank you for the suggestion of discussion on attention-based methods.
>
> ### 1. Discussion on attention-based methods
>
> Recently, attention-based methods, such as  AlphaFold and RoseTTAFoldFold, are also widely used for protein-related problems, such as protein structure prediction or generation. As we discussed in the conclusion section, attention-based techniques may potentially improve our method.
>
> Using deep learning to model a kind of data can be divided into two parts: 1) searching related elements (e.g., amino acids) and 2) encoding the local structure of the related elements.  Attention-based methods can adaptively reflect the correlation degree of two elements and thus are good at searching related elements.
> Our CDConv focuses on encoding the region of the related elements by making the most of the dual discrete and continuous nature of the data.  The two kinds of methods may facilitate each other, which can be studied in the future.
>
> In AlphaFold and RoseTTAFoldFold, there are a Multiple Sequence Alignment (MSA) branch and a pair representation branch, which interact with each other via attention mechanisms. We tried to employ the RoseTTAFoldFold model for protein classification. However, in the classification setting, MSA is not available.  Then, we attempted two solutions: 1) replace MSA with the embedding of the sequence of amino acids and 2) directly remove the MSA branch.
>
> However, neither of the two solutions achieves promising results.  For example, on superfamily fold classification, the two solutions achieved only 24.3% and 25.6%, indicating removing the branch even achieves better results (which is a bit weird). Because the RoseTTAFoldFold model is significantly modified and the network architecture may be not properly tuned, we feel it is inappropriate to compare our method with the two RoseTTAFoldFold varants based on the current results.  If you have any suggestions, please let us know.
>
>
> **We added “Appendix O: Discussion on Attention-based Methods”  in the revision.**
>
>
>
> ### 2. Terminology
>
> Thank you for pointing out the problem.
>
> The reason that we use (3+1)D is that the 3D geometry and 1D sequence structures are different: 3D is irregular and continuous but 1D is regular and discrete.  In the revision, we remove most of the (3+1)D terminology. For example, we remove the paper's time from "Continuous-Discrete Convolution for (3+1)D Geometry-Sequence Modeling in Proteins" to "Continuous-Discrete Convolution for Geometry-Sequence Modeling in Proteins".
>
> However, when describing CDConv, we still use the terminology. This is because we define ($i+j$)D CDConv as the convolution that is applied to a hybrid space with a $i$D continuous part and a  $j$D discrete part.  Using the sum cannot reflect this information because a 4D CDConv can be explained as (1+3)D, (2+2)D or (3+1)D.
>
>
> ### 3. Potential Failure Situation
>
> Our network employs a pyramid architecture, in which amino acids are downsampled. Specifically, we perform three downsamplings with rate 2. In this case, the final layer only processes $N/8$ amino acids, where $N$ is the number of amino acids.
> In this case, when a protein has only a few amino acids, e.g., less than 8, CDConv may not properly capture its structure.
>
> However, most proteins have enough amino acids for our framework. In experiments, proteins have at least 18 amino acids.  Therefore, our CDConv can correctly process them. Moreover, when proteins have too few amino acids, we can alleviate the influence of downsampling by repeating peptide chains.

---

> > ### Comment · Reviewer_hLLm · 2022-11-13
> > **Rebuttal Reply**
> >
> > Thank you for this response.  I have some follow ups re-the AlphaFold / RosettaFold comparison.
> >
> > Can you elaborate on the situations when an MSA cannot be constructed for the task of interest?  Certainly MSAs may be constructed for many multi-mer prediction problems (see e.g. how this was done in the RosettaFold and AlphaFold Multimer papers).
> >
> > And if MSAs are not truly not an option, I think the relevant baseline is to provide just a single sequence in place of the MSA (without any retraining).  This can be surprisingly effective in some cases (e.g. with RosettaFold for de novo designed proteins: https://www.nature.com/articles/s41592-021-01360-8, and the same is true with AlphaFold: see figure 4 of https://www.nature.com/articles/s41587-022-01432-w).  I think this could form the appropriate comparison (ideally with both AF and RF).
> >
> > I also do not follow the suggested division into "1) searching related elements (e.g., amino acids) and 2) encoding the local structure of the related elements".  Does the inclusion of positional encodings in attention mechanism, of sequence distance pair features into computation of attention biases (as in the IPA mechanism of AF) allow for encoding of local structure?  IIRC, RF and AF do not use explicit convolutions (only attention) but still manage to encode local structure. Can you clarify?

---

> > > ### Author Response · Authors · 2022-11-13
> > > **More Discussions about Attention-based Methods**
> > >
> > > We thank you for your quick reply.
> > >
> > > ### 1. Experiment Setting
> > >
> > > When we mentioned "in the classification setting, MSA is not available", we mean we follow the setting of exising IEConv[1,2] and GearNet [3] works. In those works, the protein classification task is treated as a standard machine learning and pattern recognition problem: given the limited and fixed traning data,  the goal is to achieve the best results on the test data.  In this way, different methods can be compared fairly. However, when using MSA, external data source or knowledge is inevitably added, which may make comparasions unfair.
> > >
> > > [1] Intrinsic-Extrinsic Convolution and Pooling for Learning on 3D Protein Structures, ICLR 2021.
> > >
> > > [2] Contrastive Pepresentation Learning for 3D Protein Structures, arXiv 2022.
> > >
> > > [3] Protein Representation Learning by Geometric Structure Pretraining, arXiv 2022.
> > >
> > > ### 2. Baseline of Replacing MSA with a Single Sequence
> > >
> > > Thank you for pointing out the baseline and the related works. We will provide the results soon.
> > >
> > > ### 3. Explanation for the Modeling Division.
> > >
> > > (To make the explanation clear, we neglect the irrelevant details and simplify the formulations.)
> > >
> > > The attention-based methods can be simplified as
> > >
> > > $\ \ \ \ \ \ \ \ \ \ \ \ \ \ \ \ \ \ \boldsymbol{f}'_i = \sum\_{j=1}^N  ~~~ \alpha\_{ij} \times g(\boldsymbol{f}_j, \boldsymbol{\delta}\_{ij}),$
> > >
> > > where $N$ is the number of amino acids, $\alpha\_{ij}$ is the attention between $i$ and $j$, $\boldsymbol{f}_i$ denotes the feature of the $i$-th amino acid and $\boldsymbol{\delta}\_{ij}$ denotes the relative relation between $i$ and $j$, e.g., displacement,
> > >
> > > Our CDConv can be simplified as
> > >
> > > $\ \ \ \ \ \ \ \ \ \ \ \ \ \ \ \ \ \ \boldsymbol{f}'_i = \sum\_{j \in \mathrm{neighbor(i)}}  ~~~  g\_{ij}(\boldsymbol{f}_j),$
> > >
> > > where $\mathrm{neighbor(i)}$ denotes the neighbors of $i$-th amino acid. The $g\_{ij}$ is the encoding function for the regular and discrete  displacement from $i$ to $j$, which is parameterized by a group of independent learnable weights.
> > >
> > > Compared to attention-based methods, our CDConv does not have an attention mechanism. To search the related amino acids, CDConv uses a simple and rigid distance-based neighbor searching method, where the neighboring region is pre-defined and fixed. In contrast, the attention-based methods can adaptively searching related amino acids via the flexible attention weights. Therefore, the attention-based methods are good at searching related amino acids.
> > >
> > > The advantage of our method is that, when encoding the local region, CDConv makes the most of the dual discrete and continuous nature of proteins.
> > >
> > > In the future, these two kinds of methods can be combined to improve each other as follows,
> > >
> > > $\ \ \ \ \ \ \ \ \ \ \ \ \ \ \ \ \ \ \boldsymbol{f}'_i = \sum\_{j=1}^N  ~~~ \alpha\_{ij} \times g\_{ij}(\boldsymbol{f}_j).$
> > >
> > > In this way, the new operation is good at searching related amino acids and makes the most of the dual discrete and continuous nature of proteins.
> > >
> > > If this explanation is still unclear, please let us know. We look forward to hearing from you soon.

---

> > > > ### Comment · Reviewer_hLLm · 2022-11-13
> > > > **replies received**
> > > >
> > > > Thank you for your follow-ups.
> > > >
> > > > My concern about not including MSAs is that their frequent availability and utility make baselines that neglect them less meaningful and important for the task at hand.

---

> > > > > ### Author Response · Authors · 2022-11-13
> > > > > **Will Include Attention-based Methods with and without MSAs**
> > > > >
> > > > > Thank you for your suggestion and explanation.
> > > > >
> > > > > We understand your concerns and will provide the results with and without MSAs as soon as possible.

---

> > > ### Author Response · Authors · 2022-11-14
> > > **Employing Protein Structure Prediction Models for Protein Classification**
> > >
> > > Think you for your suggestion.  This part is about using RoseTTAFold for protein classification.
> > >
> > > RoseTTAFold is a protein structure prediction method. There is a difference in output between protein structure prediction and classification.
> > >
> > > 1. **Protein structure prediction.** The goal of structure prediction is to make residue-wise predictions. The output is $\mathbb{R}^{N \times C}$ or $\mathbb{R}^{N \times C \times C}$, where $N$ is the number of amino acids and $C$ is the number of prediction channels. For structure prediction, $C = 3$.
> > >
> > > 2. **Protein classification.** Classification aims to make global predictions, in which a global protein representation vector is learned, i.e., $\mathbb{R}^C$. In our method, $C=1024$ or $C=2048$. Because CNNs are usually hierarchical and pyramid, which captures structures from local to global, it is natural and easy for CNNs to learn global representations.
> > >
> > > Due to the difference in output, we need to modify RoseTTAFold for classification. Specifically, we use the outputs of the feature extractor in RoseTTAFold: MSA feature $ \in \mathbb{R}^{N \times 384}$, pair representation  $\in \mathbb{R}^{N \times N \times 288}$ and xyz $\in \mathbb{R}^{N \times 3 \times 3}$.  To generate a global representation vector based on these outputs, we first perform sum on the second dimension of the pair representation and get $\mathbb{R}^{N \times 288}$ and flatten xyz as $\mathbb{R}^{N \times 9}$.
> > > Then, we concatenate MSA feature, processed pair representation and xyz: $\mathbb{R}^{N \times (384 + 288 + 9)} = \mathbb{R}^{N \times 681}$.
> > > Third, we append a fully-connected layer with ReLU to increase the dimension of the concatenation from 681 to  1024, e.g., $\mathbb{R}^{N \times 1024}$.
> > > Last, we sum those features and generate a global representation $\mathbb{R}^{1024}$. In this way, classification can be performed.
> > > We think this is the minimal modification to using RoseTTAFold for protein classification.
> > >
> > > We use the modified RoseTTAFold for protein fold classification.  The results are shown in Table 1. When MSAs are not exploited, we follow [1] to only use the peptide chain itself. When Templates are not exploited, we use the input protein geometry structure as the only template. Probably because the architecture of RoseTTAFold is not suitable for global representation learning, it achieves limited accuracy.  Converting RoseTTAFold to a pyramid architecture may boost the accuracy for protein classification. It can be studied in the future.
> > >
> > > We notice that MSAs and protein templates help classification. They may play a data augmentation role in protein classification. Moreover, protein templates help more than MSAs. This may indicate that the 3D geometry structure is more important than the 1D sequence structure for protein recognition.
> > >
> > > [1] Deep Learning and Protein Structure Modeling. Nature Methods 2022.
> > >
> > > **Table 1:** Accuracy (\%) of the modified RoseTTAFold by us on protein fold classification.
> > >
> > > | Method | Fold | Superfamily | Family |
> > > | ---- | ---- | ---- | ----- |
> > > | GraphQA | 23.7 | 32.5 | 84.4 |
> > > | GVP | 16.0 | 22.5 | 83.8 |
> > > | IEConv (residue level) | 47.6 | 70.2 | 99.2 |
> > > | GearNet | 28.4 | 42.6 | 95.3 |
> > > | GearNet-IEConv | 42.3 | 64.1 | 99.1 |
> > > | GearNet-Edge | 44.0 | 66.7 | 99.1 |
> > > | GearNet-Edge-IEConv | 48.3 | 70.3 | 99.5 |
> > > | CDConv (ours) | 56.7 | 77.7 | 99.6 |
> > > | RoseTTAFold | 22.7 | 28.2 | 85.5 |
> > > | RoseTTAFold (with MSA) | 24.2 | 31.4 | 87.0 |
> > > | RoseTTAFold (with Template) | 31.8 | 39.5 | 91.1 |
> > > | RoseTTAFold (with MSA and Template) | 33.4 | 40.4 | 92.3 |
> > >
> > > **We added “Appendix R: Employing Protein Structure Prediction Models for Protein Classification” in the revision.**
> > >
> > > If you feel our efforts address your concerns, we would appreciate it if you could kindly consider raising your rating. Your support is significantly important to our paper. Should you need further information, please let us know. We look forward to hearing from you soon. Thank you!

---

### Official Review · Reviewer_qyBv · 2022-10-25

**Confidence:** 3
**Correctness:** 3
**Technical Novelty And Significance:** 2
**Empirical Novelty And Significance:** 2
**Recommendation:** 6

**Clarity, Quality, Novelty And Reproducibility:**

The paper is overall easily readable.

The proposed idea lacks technical depth and the method lacks important details. See the weaknesses above.

The novelty of the proposed idea is marginal.

The reproducibility is OK.

**Strength And Weaknesses:**

Strength:
1. The paper address a very important problem.
2. The paper did a good job of integrating 1D features of the peptide chain and 3D features of amino acid coordinates.
3. The rotation invariance is guaranteed by the convolution kernel design.

Weakness:
1. There is no formal problem formulation.
2. The description of the experiment is not clear. For example, the author did not provide details about how to mask the central amino acids in 4.4.


**Summary Of The Paper:**

This paper proposed a Continuous-Discrete Convolution (CDConv) for Geometry-Sequence modeling in proteins. The goal is to increase the accuracy of four tasks, protein fold classification, enzyme reaction classification, gene ontology term prediction, and enzyme commission number prediction. The paper also explained how the CDConv captures the features of protein structure.

**Summary Of The Review:**

The paper is to unify continuous convolution and discrete convolution for modeling protein structure. They claim that the results of their method are state-of-the-art accuracy in four tasks. In addition, they explained how the CDConv capture the features of protein structure, i.e., the early layers tend to focus on the central region of the protein and the late layers can capture the whole structure.

However, there are some issues in this paper:
a) The author did not give a formal problem formulation.
b) The description of the experiment is not clear. In section 4.4, they did not give the details of how to mask the central area. The mask method may change the feature of the continuous features. Thus, more accuracy may be caused by the mask method rather than the importance of central amino acids.

---

> ### Author Response · Authors · 2022-11-13
> **Problem Formulation and Experiment Details**
>
> We thank you for acknowledging that our paper **"address a very important problem"** and **"did a good job of integrating 1D features of the peptide chain and 3D features of amino acid coordinates."** We also thank you for your valuable comments.
>
>
> ### 1. Problem Formulation
>
> To evaluate the proposed CDConv, we apply it to the protein classification problem. Classification is a fundamental task for pattern recognition and representation learning.
>
> Specifically,  given a protein, which consists of a list of amino acid coordinates $\boldsymbol{P} \in \mathbb{R}^{3 \times N}$ and a list of amino acid types $\boldsymbol{F}  \in \mathbb{R}^{1 \times N}$,  where $N$ is the number of amino acids, the goal of protein classification is to map $\boldsymbol{P}$ and $\boldsymbol{F}$ to a prediction vector $\boldsymbol{y} \in \mathbb{R}^k$, where $k$ is the number of protein classes or categories,
>
> $\ \ \ \ \ \ \ \ \ \ \ \ \ \ \ \ \ \ \ \ \ \ \ \ \ \ \ \ \ \ \ \ \ \ \boldsymbol{y} = \Phi(\boldsymbol{P}, \boldsymbol{F}; \Theta).$
>
> The $\Phi$ is a deep neural network and $\Theta$ is its parameters.
>
> Then, for single-label classification, the softmax function is usually used to convert $\boldsymbol{y}$ to prediction probabilities, i.e., $softmax(\boldsymbol{y})$. The class with the highest probability will be considered as the final prediction.
>
> For multi-label classification, the sigmoid function is usually used to convert $\boldsymbol{y}$ to binary prediction probabilities, i.e., $\sigma(\boldsymbol{y})$.
> Those classes whose probabilities are higher than a predefined threshold, e.g., 0.5,  will be considered as the final predictions.
>
> **We added “Appendix A: Problem Formulation”  in the revision.**
>
> ### 2. More Details of the Study on the Importance of Surface and Central Amino Acids
>
> To investigate which of the surface and central regions is more important, we mask the same number (20%) of amino acids in the surface and central areas, respectively.
>
>
> Specifically, we first calculate the center of a protein as follows,
>
> $\ \ \ \ \ \ \ \ \ \ \ \ \ \ \ \ \ \ \ \ \ \ \bar{\boldsymbol{p}} = \frac{1}{N} \sum_{i=1}^N \boldsymbol{p}_i$,
>
> where $N$ is the number of amino acids and $\boldsymbol{p}_i$ denotes the 3D coordinate of the $i$-th amino acid.
> For surface region masking, we remove the 20% farthest amino acids from the center based on the geometry distance.
> For central region masking, we remove the 20% closest amino acids to the center.
>
>
> ### 3. Novelty
>
> In 2012, the appearance of the AlexNet Convolutional Neural Network (CNN) caused a revolution in machine learning, which significantly boosted the accuracy of ImageNet classification.  Since then, many CNNs are proposed for image classification, such as VGG, Inception, ResNet, DenseNet, etc. At the same time, Multilayer Perceptrons (MLPs) and Recurrent Neural Networks (RNNs, e.g., LSTM and GRU).  Recently, Graph Neural Networks (GNNs) and Transformers are also proposed to model various types of data. In summary, MLP, CNN, RNN, GNN and Transformer are the most fundamental tools in deep learning. Most deep neural networks are based on the five tools.
>
> In the early years, most CNNs, which are mainly used to process regular images and natural languages, are discrete. In recent years, to capture the structure of irregular point clouds for the 3D perception of self-driving cars and robots, continuous CNNs are widely proposed.
>
> In our paper, we propose a new class of CNNs, i.e., CDConv, which can simultaneously process regular and irregular data and make the most of the dual discrete and continuous nature of the data. We think our CDConv is novel because it is the first time to unify discrete and continuous CNNs.  Actually, existing discrete CNN and continuous CNNs can be seen as special cases of our CDConv.  Therefore, this paper may make a certain contribution to one of the five fundamental deep learning tools.

---

### Official Review · Reviewer_nEe2 · 2022-10-26

**Confidence:** 2
**Correctness:** 3
**Technical Novelty And Significance:** 3
**Empirical Novelty And Significance:** 3
**Recommendation:** 6

**Clarity, Quality, Novelty And Reproducibility:**

**Minor Questions/Problems:**

Figure 3: according to Figure 3(b), it seems t+1 is also a valid neighbor within the dashed ball (<r) and it is close to t in the sequence (<l/2). Why is it not considered a neighbor of t in the caption?

**Strength And Weaknesses:**

**Strengths:**

1. The motivation is clearly explained.

2. The proposed framework is original and powerful (in experiments).

3. The framework suits the application of protein representation learning well, and it can potentially inspire the community for representing other complex entities.

**Weaknesses:**

1. It seems the embedding method can only be applied to specific tasks. As stated in Section 3.4, the convolution design satisfies rotation invariant, which is sufficient for properties prediction. However, rotation equivariance is usually required for broader scenarios of protein representation. It is questionable whether the proposed scheme can still fit in.

2. The computational efficiency and the scalability of the proposed method should have been discussed. Considering the protein sequence can go more than thousands of tokens, and the current convolution scheme stack multiple layers to progressively enlarge its receptive field, it is necessary to investigate its performance (and speed) on relatively large proteins.

3. As the experimental results are very impressive, revealing the program really helps back up the reliability of the reported scores.


**Summary Of The Paper:**

This work proposed a new genre of convolution, namely continuous-discrete convolution (CDConv) for protein representation learning, which is folded from 1D discrete amino acid chains into a 3D continuous space. Rather than separately aggregating 1D and 3D representation, the proposed (3+1)D CDConv unifies continuous and discrete convolutions in the 3D and 1D spaces. The proposed method was validated on a variety of protein properties prediction (classification) tasks and achieves SOTA performance on them.

**Summary Of The Review:**

The paper has proposed an interesting convolution model for protein representation learning which is of good quality.

---

> ### Author Response · Authors · 2022-11-13
> **Part I: Rotation Equivariance for Other Scenarios**
>
> We thank you for acknowledging our work **"is original and powerful"**, **“has proposed an interesting convolution model”** and **"can potentially inspire the community for representing other complex entities."** We also thank you for your constructive comments.
>
> ### 1. Rotation equivariance for other scenarios
>
> In this paper, we propose a new class of convolution, i.e., CDConv, to make the most of the dual discrete and continuous nature of
> the data to avoid the impact of regular and irregular modeling on each other. The key design of CDConv is to employ
> independent learnable weights for different regular displacements but directly encode irregular displacements. As a prototype, CDConv can be instantiated to various instances.  In this paper, we implement a (3+1)D CDConv for protein representation learning. When implementing the 3D geometry modeling part, we use the relative spatial encoding technique.
>
> The relative spatial encoding is independent of our CDConv and can be replaced with other techniques. When for other scenarios, we can integrate the corresponding rotation equivariance technique into CDConv accordingly. Moreover, there can be a low-bound solution to completely avoid the rotational variance problem for our method: only use the distance information and do not model direction,
>
> $\ \ \ \ \ \ \ \ \ \ \ \ \ \ \ \ \ \ \ \ \ \ \ \ \ \ \ \ \ \ \ g\big(\mathbf{p}_{t+ \Delta} - \mathbf{p}_t; \mathbf{\theta}\_{\Delta} \big) = \mathbf{\theta}\_{\Delta}  \cdot (||\mathbf{p}\_{t+\Delta} - \mathbf{p}_t||/r),$
>
> where the distance is normalized with the neighbor searching radius $r$.
>
> Then, we evaluate this distance-only-based "completely-rotationally-invariant" (3+1)D CDConv on protein fold classification. As shown in Table 1, the new (3+1)D CDConv still achieves comparable accuracy with the previous state-of-the-art methods.
>
> **Table 1:** Accuracy (%) of distance-only-based "completely-rotationally-invariant" (3+1)D CDConv on protein fold classification.
>
> | Method | Fold | Superfamily | Family |
> | ---- | ---- | ---- | ----- |
> | GraphQA | 23.7 | 32.5 | 84.4 |
> | GVP | 16.0 | 22.5 | 83.8 |
> | IEConv (residue level) | 47.6 | 70.2 | 99.2 |
> | GearNet | 28.4 | 42.6 | 95.3 |
> | GearNet-IEConv | 42.3 | 64.1 | 99.1 |
> | GearNet-Edge | 44.0 | 66.7 | 99.1 |
> | GearNet-Edge-IEConv | 48.3 | 70.3 | 99.5 |
> | Distance-only-based CDConv w/o distance normalization (ours) | 46.6 | 73.2 | 99.3 |
> | Distance-only-based CDConv w/ distance normalization (ours) | 48.6 | 73.5 | 99.4 |
> | Relative-spatial-encoding-based CDConv (ours) | 56.7 | 77.7 | 99.6 |
>
> **We added “Appendix M: Rotation Equivariance for Other Scenarios” in the revision.**

---

> > ### Author Response · Authors · 2022-11-13
> > **Part II: Computational Efficiency and Scalability**
> >
> > ### 2. Computational efficiency and scalability
> >
> > Thank you for the question.
> >
> > First, we would like to highlight that our method is a Convolutional-Neural-Network-(CNN)-style approach. The architectures or frameworks of CNNs are usually hierarchical or pyramid, in which data is exponentially downsampled as networks deepen. This is one of the reasons that CNN sometimes can be very deep, such as ResNet-152, which consists of 152 layers.
> >
> > Second, our CDConv architecture only contains 8 layers and we downsample amino acids every two layers. In this case, our method is able to process relatively large proteins. Specifically, in experiments, we found there are many proteins that have more than 2000 amino acids and our method can still handle them even with batch 64.
> >
> > To investigate the performance of our method on relatively large proteins, we collect the proteins that contain more than 500 amino acids from  the enzyme reaction classification dataset, leading to the following sub-sets (Table 2),
> >
> > **Table 2:** Details of the subset of large proteins for enzyme reaction classification.
> >
> > | Split | Largest Protein | Smallest Protein | Average Number of Amino Acids per Protein | Dataset Size |
> > | ---- | ---- | ---- | ---- | ---- |
> > | Training | 3,725 | 500 | 722.3 |2,498|
> > | Validation | 1,314 | 500 | 711.6 |319|
> > | Test | 1,119 | 500 | 611.0 | 450 |
> >
> >
> > where the original dataset is as follows (Table 3),
> >
> > **Table 3:** Details of the original dataset for enzyme reaction classification.
> >
> >
> > | Split | Largest Protein | Smallest Protein | Average Number of Amino Acids per Protein | Dataset Size |
> > | ---- | ---- | ---- | ---- | ---- |
> > | Training | 3,725 | 18 | 296.2 | 29,215 |
> > | Validation | 1,314 | 24 | 346.0 | 2,562 |
> > | Test | 1,119 | 20 | 293.1 | 5,651 |
> >
> >
> > However, because the training data significantly reduces (from 29,215 to 2,498 proteins), the subset of large proteins may lead to inferior accuracy. Therefore, for a fair comparison, we randomly sample a subset that contains the same number of proteins as that of large proteins. The details are as follows (Table 4).
> >
> >
> > **Table 4:** Details of the normal subset of randomly sampled proteins for enzyme reaction classification.
> >
> > | Split | Largest Protein | Smallest Protein | Average Number of Amino Acids per Protein | Dataset Size |
> > | ---- | ---- | ---- | ---- | ---- |
> > | Training | 2,650 | 23 | 293.6 | 2,498 |
> > | Validation | 1,314 | 31 | 363.1 | 319 |
> > | Test | 1,119 | 25 | 299.3 | 450 |
> >
> >
> >
> >
> > Then, we conduct enzyme reaction classification on the two subsets with a single Nvidia Quadro RTX A5000 GPU and Intel(R) Xeon(R) Silver 4214R CPU @ 2.40GHz. We also report the mean running time per prediction. As shown in Table 5, our CDConv can still effectively and efficiently recognize enzyme reactions.
> >
> > **Table 5:** Performance comparison of CDConv on enzyme reaction classification between large and normal proteins.
> >
> > | Subset  | Accuracy | Running Time |
> > | ---- | ---- | ---- |
> > | Normal Proteins | 67.7% | 10.2 ms |
> > | Large Proteins | 65.3% | 13.2 ms |
> >
> > **We added “Appendix N: Computational Efficiency and Scalability”  in the revision.**
> >
> >
> > ### 3. Code Release
> >
> > Our code will be publicly available.
> >
> > ### 4. Clarity
> >
> > “Figure 3: according to Figure 3(b), it seems t+1 is also a valid neighbor within the dashed ball (<r) and it is close to t in the sequence (<l/2). Why is it not considered a neighbor of t in the caption?”
> >
> > We thank you for pointing out the problem. We corrected it in the revision.

---

### Author Response · Authors · 2022-11-13
**Summary of Revision**

Dear Chairs and Reviewers,


We would like to thank the reviewers for their careful and constructive comments. We also thank the reviewers for acknowledging our work is original, powerful, effective (nEe2,hLLm), novel (JFLd) and addresses a very important problem (qyBv). The paper has been revised in accordance with the reviewers’ comments and suggestions. Updates and changes are marked by **red** color in the revised version.  The major changes in this revision lie in nine aspects:

1. Appendix A: Problem Formulation.
2. Appendix J: More Details and Results of the Study on the Importance of Surface and Central Amino Acids.
3. Appendix M: Rotation Equivariance for Other Scenarios.
4. Appendix N: Computational Efficiency and Scalability.
5. Appendix O: Discussion on Attention-based Methods.
6. Appendix P: Influence of Missing Coordinates.
7. Appendix Q: Influence of Voxel Size for Discretizing 3D Coordinates.
8. Appendix R: Employing Protein Structure Prediction Models for Protein Classification. (This is the first attempt to explore using protein structure prediction models for protein classification.)
9. Appendix S: Generalized CDConv for Explicitly Dealing with Missing 3D Coordinates. (This is the first model to deal with missing 3D coordinates for protein classification or recognition.)



Should you need further information, please let us know. We look forward to hearing from you soon.

Yours sincerely,

Authors of Paper “Continuous-Discrete Convolution for Geometry-Sequence Modeling in Proteins”

---

### Decision · Program_Chairs · 2023-01-20

**Decision:**

Accept: poster

**Justification For Why Not Higher Score:**

This is a good paper that yields convincing empirical results, but the novelty is not ground breaking enough to warrant a spotlight/oral presentation.

**Justification For Why Not Lower Score:**

The AC and reviewers all agree that this is a good paper, of significant interest and with interesting results. We see no reason for rejecting the paper.

**Metareview: Summary, Strengths And Weaknesses:**

The paper combines discrete and continuous convolution to jointly model protein sequence and structure and demonstrates the effectiveness of their approach on various tasks.

Strength:
All reviewers and AC agree that this paper addresses an important problem, that the method is well motivated and sound.

We congratulate the authors on their additional work to address the reviewer points. The significance of the paper is greatly enhanced by the newly added material on attention methods, computational efficiency, robustness w.r.t. missing coordinates. etc.

Weaknesses: We strongly encourage the authors to investigate whether the central region of the protein is really more important or whether it is an artifact of the approach. Also it would be important to instigate the possible failure modes of downsampling.

Minor comment: In the appendix T, "No matte" --> "No matter"



**Note From Pc:**

if the above contains the word "oral" or "spotlight" please see: "oral" presentation means -> notable-top-5% and "spotlight" means -> notable-top-25%. As stated in our emails, we are disassociating presentation type from AC recommendations